# LOSS TRANSFORMATION INVARIANCE IN THE DAMPED NEWTON METHOD

## ABSTRACT

The Newton method is one of the most widely used second-order optimization techniques, valued for its conceptual simplicity and extremely fast local convergence. A key advantage is its invariance under affine transformations (e.g., choice of coordinate basis), which greatly facilitates implementation. However, the classical Newton method fails to converge when initialized far from the solution, motivating the development of various globalization techniques. In this work, we focus on stepsize damping, which, when appropriately scheduled, ensures fast global convergence while preserving both affine-invariance and superlinear local rates. Although highly effective in convex settings, existing algorithms offer limited guarantees for problems that are only nearly convex. To address this, we investigate loss transformations that convexify the objective. We show that Newton stepsize schedules are invariant under such transformations and that stepsize scheduling implicitly searches over the space of objective transformations. Our theoretical findings are further supported by comprehensive experimental validation.

## 1 INTRODUCTION

The Newton method is one of the most fundamental algorithms in optimization. Its origins trace back to the works of Newton (1687) and Raphson (1697), and its numerous variants have found applications across mathematics, statistics, and machine learning. For example, the survey of trust-region and quasi-Newton methods by Conn et al. (2000) cites over a thousand related papers. The method seeks a minimizer of a smooth twice differentiable function $f : \mathbb{R}^d \to \mathbb{R}$ via the update rule

$$x^{k+1} = x^k - \left[\nabla^2 f(x^k)\right]^{-1} \nabla f(x^k). \tag{1}$$

It is widely appreciated for its extremely fast local convergence and for its affine invariance, which ensures robustness to scaling and changes of coordinate basis.

Despite its rich history, even the most basic variants of the Newton methods – stepsize schedules – are still being researched to this day. Nesterov & Nemirovski (1994) proposed one of the first stepsize schedules with global convergence guarantees. More recently, Hanzely et al. (2022) established a duality between Newton stepsize schedules and Levenberg–Marquardt regularization, achieving a global $\mathcal{O}(1/k^2)$ rate. Hanzely et al. (2024) extended this analysis under assumptions similar to third-order tensor methods, obtaining a simple schedule with $\mathcal{O}(1/k^3)$ convergence.

When the loss function has unknown parameterization, explicit stepsize schedules are often replaced by linesearch procedures, including greedy rules or backtracking based on standard descent conditions such as Frank–Wolfe, strong Frank–Wolfe, Armijo, or Goldstein conditions. For Newton direction, these procedures typically select stepsizes from the range $(0, 1]$ (Nocedal & Wright, 2006).

All of these methods, however, fundamentally rely on convexity. In fact, without convexity the direction $-\left[\nabla^2 f(x^k)\right]^{-1} \nabla f(x^k)$ is not necessarily a descent direction, and the stepsized Newton methods can diverge, both in theory and in practice. Moreover, explicit stepsize rules (Nesterov & Nemirovski, 1994; Hanzely et al., 2022; 2024) rely on the *Newton decrement* $\left\langle \nabla f(x^k), \left[\nabla^2 f(x^k)\right]^{-1} \nabla f(x^k) \right\rangle$, which is not necessarily positive outside convexity. Another drawback of nonconvex analysis is viewpoint that the Newton linesearches can be interpreted as Hessian-regularized updates (Hanzely et al., 2024):

$$x^{k+1} = x^k - \left[\nabla^2 f(x^k) + \lambda_k \mathbf{I}\right]^{-1} \nabla f(x^k), \tag{2}$$

with $\lambda_k \propto \|x^{k+1} - x^k\|^\beta$ or $\lambda_k \propto \|\nabla f(x^k)\|^\beta$. In convex settings, this regularization admits a natural interpretation, since $\lambda_k$ can be related to the Newton decrement, yielding an equivalent update along the Newton direction. However, for nonconvex functions, the analogous regularization based on Newton decrement would be negative, making the analogy invalid. This raises a fundamental question:

> *Is convexity truly a necessary condition for the convergence of Newton's method, or can one guarantee convergence in the nonconvex regime?*

In this work we show that the standard form of convexity is not required for the stepsized Newton method to converge. We prove that Newton's stepsize schedules are *invariant under loss transformations*, enabling convexification of objectives without altering the iterates themselves. This new property, which we call *transformation invariance*, extends the known geometric invariances of Newton's method.

### 1.1 SUMMARY OF THE CONTRIBUTIONS

1. **Conceptual advance:** We introduce the notion of *transformation invariance* and prove that the stepsized Newton method enjoys this property (Theorem 1). In contrast, closely related Hessian regularization techniques do not (Lemma 2).

2. **Theoretical consequences:** Transformation invariance enables convexification (Theorem 3) and star-convexification (Theorem 4) of pseudoconvex losses. We provide sufficient and necessary conditions for such transformations to exist (Theorem 3).

3. **Practical consequences:** We propose a *transformation-induced stepsize schedule* (Corollary 1), which transfers the iterate sequence of the Newton method applied to a transformed loss back to the original objective.

4. **Explaining non-standard stepsizes:** Our framework provides a principled explanation for the effectiveness of unconventional stepsize ranges in Newton's method, including stepsizes larger than one (Section 2.1, Section 2.2) and even negative stepsizes (Figures 2, 4).

### 1.2 PRELIMINARIES

We consider the unconstrained minimization problem

$$\min_{x \in \mathbb{R}^d} f(x), \tag{3}$$

where $f : \mathbb{R}^d \to \mathbb{R}$ is a twice continuously differentiable function, and $x_* \overset{\text{def}}{=} \operatorname{argmin}_{x \in \mathbb{R}^d} f(x)$ denotes its global minimizer. We denote $\|\cdot\|$ for the standard Euclidean norm, and we will also be using quadratic forms induced by symmetric matrices $\mathbf{B} \in \mathbb{R}^{d \times d}$ and their pseudoinverses,

$$\|h\|_{\mathbf{B}}^2 \overset{\text{def}}{=} \langle h, \mathbf{B}h \rangle, \qquad \|g\|_{\mathbf{B}}^{*2} = \left\langle g, \mathbf{B}^\dagger g \right\rangle, \qquad \forall h, g \in \mathbb{R}^d.$$

In particular, we frequently use local Hessian quadratic forms, $\mathbf{B} = \nabla^2 f(x)$, for which we use the shorthand

$$\|h\|_x^2 \overset{\text{def}}{=} \left\langle h, \nabla^2 f(x)h \right\rangle, \qquad \|g\|_x^{*2} \overset{\text{def}}{=} \left\langle g, \nabla^2 f(x)^\dagger g \right\rangle.$$

Throughout the paper, we analyze properties of convex and pseudoconvex functions.

**Definition 1** (Convexity). *Function $f : \mathbb{R}^d \to \mathbb{R}$ is called convex, if*
$$f(y) \geq f(x) + \langle \nabla f(x), y - x \rangle, \quad \forall x, y \in \mathbb{R}^d.$$

We relax the standard convexity assumption to the following notion of pseudoconvexity.

**Definition 2** (Pseudoconvexity). *Function $f : \mathbb{R}^d \to \mathbb{R}$ is called pseudoconvex, if*
$$f(y) < f(x) \Rightarrow \langle \nabla f(x), y - x \rangle < 0, \quad \forall x, y \in \mathbb{R}^d.$$
*Additionally, $f$ is called strictly pseudoconvex, if*
$$f(y) \leq f(x) \Rightarrow \langle \nabla f(x), y - x \rangle < 0, \quad \forall x \neq y \in \mathbb{R}^d.$$

This formulation highlights the close connection between convexity and pseudoconvexity. For our analysis, we rely on an equivalent characterization from Avriel & Schaible (1978).

**Lemma 1.** *Function $f$ is (strictly) pseudoconvex if and only if the following conditions hold:*

1. *$v^T \nabla f(x) = 0 \Rightarrow v^T \nabla^2 f(x) v \geq 0$,*

2. *If $\nabla f(x) = 0$, then $x$ is a (strict) global minimum.*

## 2 LOSS TRANSFORMATIONS

In this section, we analyze loss transformations. Consider a mapping $\phi : \mathbb{R} \to \mathbb{R}$ and the composite function $L \overset{\text{def}}{=} \phi \circ f$. Instead of minimizing the original, difficult nonconvex objective $f$, we may seek a transformation $\phi^1$ such that the transformed loss $L = \phi \circ f$ is convex and easier to optimize. By the chain rule, the gradient and Hessian of $L$ are

$$\nabla L(x) = \phi'(f(x)) \nabla f(x), \tag{4}$$

$$\nabla^2 L(x) = \phi'(f(x)) \nabla^2 f(x) + \phi''(f(x)) \nabla f(x) \nabla f(x)^\top. \tag{5}$$

Rearranging, we observe that $\nabla^2 L(x) \propto \nabla^2 f(x) + \frac{\phi''(f(x))}{\phi'(f(x))} \nabla f(x) \nabla f(x)^\top$, which suggests that $L$ can become convex if $\phi$ is chosen appropriately. A natural candidate is the exponential transformation $\phi(f(x)) = \exp(a \cdot f(x))$, in which case the Hessian simplifies to $\nabla^2 L(x) \propto \nabla^2 f(x) + a \nabla f(x) \nabla f(x)^\top$.

Thus, a carefully selected transformation can substantially improve the optimization landscape. Once convexified, the transformed objective $L$ can be efficiently minimized using the Newton method,

$$x^+ = x - \alpha_L \left[ \nabla^2 L(x) \right]^{-1} \nabla L(x). \tag{6}$$

### 2.1 EQUIVALENCE OF THE NEWTON'S METHOD

Surprisingly, applying the stepsized Newton method to the original loss $f$ or to its transformed version $L = \phi \circ f$ yields almost identical behavior. The descent directions coincide, and the only difference lies in the stepsize. This observation is formalized in the following theorem.

---

**Theorem 1.** *Let $\alpha(x)$ be an arbitrary Newton stepsize schedule for the original loss $f(x)$. If $\phi$ is strictly monotone, $\nabla f(x) \in \text{Range}\left(\nabla^2 f(x)\right)$, and $\phi'(f(x)) + \phi''(f(x)) \|\nabla f(x)\|_x^{*2} \neq 0$, then the Newton method on the transformed loss $L(x)$ with stepsize*

$$\alpha_L(x) \overset{\text{def}}{=} \alpha(x) \left( 1 + \frac{\phi''(f(x))}{\phi'(f(x))} \|\nabla f(x)\|_x^{*2} \right) \tag{7}$$

*produces the same sequence of iterates as the Newton method on $f(x)$ with stepsize $\alpha(x)$, i.e.,*

$$x - \alpha(x) \left[ \nabla^2 f(x) \right]^\dagger \nabla f(x) = x - \alpha_L(x) \left[ \nabla^2 L(x) \right]^\dagger \nabla L(x). \tag{8}$$

*We refer to this property as the* transformation invariance *of the stepsized Newton method. The multiplicative term $1 + \frac{\phi''(f(x))}{\phi'(f(x))} \|\nabla f(x)\|_x^{*2}$ is called the* scaling factor.

---

**Corollary 1.** *A Newton method on the original function $f$ with stepsize schedule $\alpha(x) \overset{\text{def}}{=} \alpha_L(x) \left( 1 + \frac{\phi''(f(x))}{\phi'(f(x))} \|\nabla f(x)\|_x^{*2} \right)^{-1}$ produces the identical sequence of iterates to the Newton method on the transformed loss $L$ with stepsize schedule $\alpha_L(x)$.*

We illustrate the importance of this result in Figure 1 (see also Section 4.3 for details).

Vanilla and damped Newton methods, which rely on the Hessian inverse, suffer from ill-defined Hessians; thus, convergence is typically analyzed only in convex or strongly convex settings, leaving nonconvex cases uncovered. In practice, Newton iterations are more stably performed via the MINRES method (Paige & Saunders, 1975), based on a least-squares formulation using the pseudoinverse (Roosta et al., 2018; Fong & Saunders, 2012), rather than conjugate gradients (Nocedal & Wright, 2006). Considering the Hessian pseudoinverse broadens the class of problems where the damped Newton method applies. When the Hessian is invertible, $\text{Range}(\nabla^2 f(x)) = \mathbb{R}^d$, so $\nabla f(x) \in \text{Range}\left(\nabla^2 f(x)\right)$ holds automatically, which justifies this assumption in Theorem 1.

---

[1]The transformation $\phi$ must be monotonically increasing on $[f(x_*), \infty)$ to preserve the minimizer, i.e., $\text{argmin}_{x \in \mathbb{R}^d} L(x) = x_*$.

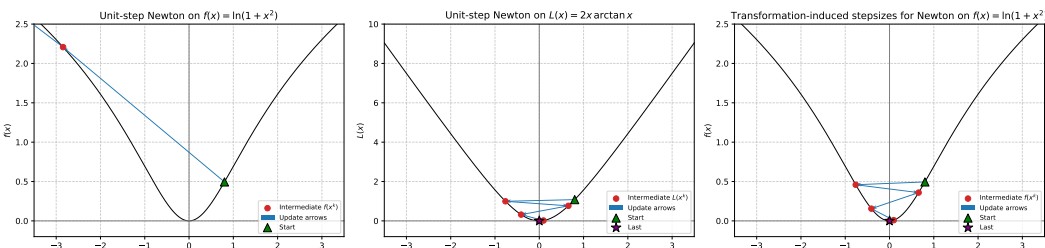

Figure 1: **Toy example: effect of transformation-induced stepsizes.** Loss $f(x) = \ln(1 + x^2)$ initialized at $x_0 = 0.8$. Left: the classical Newton method diverges. Middle: Newton method on the surrogate $L(x) = 2x \arctan(x)$ converges to the minimizer. Right: Newton method on the original loss with a transformation-induced stepsize schedule also converges.

Despite the known connection between Newton stepsizes and Levenberg–Marquardt Hessian regularization (Hanzely et al., 2022), transformation invariance does not extend to the standard Levenberg–Marquardt scheme. This limitation is formalized below.

**Lemma 2.** *Consider the sequence of iterates generated by the Newton method with a Levenberg–Marquardt regularization schedule $\lambda(x)$ applied to $f(x)$. There exists **no** regularization schedule $\lambda_\phi(x, \phi(\cdot), \lambda(x))$ for $\phi(f(x))$ producing the identical sequence of iterates.*

As a result, loss invariance, established for damped Newton method, is not valid for Levenberg–Marquardt regularization, which includes quadratic regularization with gradient norms (Doikov & Nesterov, 2024), regularized Newton method (Mishchenko, 2023), super-universal Newton method (Doikov et al., 2024a).

Next, we turn to the geometric properties of the transformation $\phi$ and provide motivating examples.

To preserve the minimizer, we restrict $\phi$ to be monotonically increasing on the range of $f(x)$. Since $\phi$ preserves contour lines, a necessary condition for $\phi \circ f$ to be convex is that the sublevel sets of $f$ are themselves convex.

The stepsize scaling factor $1 + \frac{\phi''(f(x))}{\phi'(f(x))} \|\nabla f(x)\|_x^{*2}$ is invariant under rescaling or translation of $\phi$: for linear transformations of the loss, the scaling factor is exactly one. A larger scaling factor implies that the transformed loss admits larger stepsizes, while a negative factor allows negative stepsizes (Theorem 1). Importantly, in all cases the sequences of iterates remain identical. Thus, if the stepsized Newton method converges on the original loss, the transformation-induced schedule guarantees convergence as well.

Having established these basic geometric properties of $\phi$, we now turn to concrete motivating examples.

EXAMPLE: POLYNOMIAL FUNCTION

It is well known that the classical Newton method minimizes a quadratic function in a single step. A natural question is whether this property extends to higher-order polynomials. Theorem 1 suggests that the answer is indeed positive. For a positive definite matrix $\boldsymbol{A} \succ \boldsymbol{0}$ and an exponent $p \neq 1$, denote

$$f(x) = \frac{1}{p}\|x\|_{\boldsymbol{A}}^p, \qquad \nabla f(x) = \|x\|_{\boldsymbol{A}}^{p-2}\boldsymbol{A}x, \qquad \nabla^2 f(x) = (p-2)\|x\|_{\boldsymbol{A}}^{p-4}\boldsymbol{A}xx^\top\boldsymbol{A} + \|x\|_{\boldsymbol{A}}^{p-2}\boldsymbol{A}. \tag{9}$$

Running Newton method on $f$ yields update

$$x^+ = x - \alpha \left[\nabla^2 f(x)\right]^{-1} \nabla f(x) \tag{10}$$

$$= x - \alpha \left[(p-2)\|x\|_{\boldsymbol{A}}^{p-4}\boldsymbol{A}xx^\top\boldsymbol{A} + \|x\|_{\boldsymbol{A}}^{p-2}\boldsymbol{A}\right]^{-1} \|x\|_{\boldsymbol{A}}^{p-2}\boldsymbol{A}x \tag{11}$$

$$= \left(\mathbf{I} - \alpha \left[(p-2)\|x\|_{\boldsymbol{A}}^{-2}xx^\top\boldsymbol{A} + \mathbf{I}\right]^{-1}\right)x, \tag{12}$$

Table 1: Comparison of loss transformations and corresponding stepsize compensation factor.

| Loss transformation | Loss transformation formula $\phi(f(x))$ | Stepsize scaling factor |
|---|---|---|
| Linear | $a \cdot f(x) + b$ | $1$ |
| Polynomial | $f(x)^r$ | $1 + \frac{(r-1)}{f(x)}\|\nabla f(x)\|_x^{*2}$ |
| Exponential | $\exp(a \cdot f(x))$ | $1 + a\|\nabla f(x)\|_x^{*2}$ |
| Logarithmic | $\log(a + f(x))$ | $1 - \frac{1}{a+f(x)}\|\nabla f(x)\|_x^{*2}$ |
| Sigmoid | $\sigma(f(x)) = (1 + \exp(-f(x)))^{-1}$ | $1 + (1 - 2\sigma(f(x)))\|\nabla f(x)\|_x^{*2}$ |

and because $\left[(p-2)\|x\|_{\boldsymbol{A}}^{-2}xx^\top \boldsymbol{A} + \mathbf{I}\right] x = (p-1)x$, so $\left[(p-2)\|x\|_{\boldsymbol{A}}^{-2}xx^\top \boldsymbol{A} + \mathbf{I}\right]^{-1} x = \frac{1}{p-1}x$ for $p \neq 1$, we can conclude

$$= \left(1 - \frac{\alpha}{p-1}\right) x. \tag{13}$$

Therefore, stepsize $\alpha = p - 1$ guarantees convergence in 1 iteration.

## 2.2 EXAMPLE: POLYTOPE FEASIBILITY

Polytope feasibility problem searches for a point from a polytope $\left\{x \in \mathbb{R}^d | \langle a_i, x \rangle \le b_i, \ \forall i \in \{1, \ldots, n\}\right\}$, and can be reformulated with $(t)_+ \stackrel{\text{def}}{=} \max\{t, 0\}$ and $p \ge 2$ as

$$\min_{x \in \mathbb{R}^d} \left\{f_p(x) \stackrel{\text{def}}{=} \sum_{i=1}^n (\langle a_i, x \rangle - b_i)_+^p\right\}. \tag{14}$$

Hanzely et al. (2022) observed that for this problem the optimal fixed stepsizes for the Newton method are approximately $0.95, 1.95, 2.95, 3.95$ for $p = 2, 3, 4, 5$, respectively. Our theory provides a natural explanation for this phenomenon via the approximation $f_p(x) \approx (f_2(x))^{p/2}$.

## 3 TRANSFORMATIONS IMPROVING OBJECTIVE PROPERTIES

The Newton method and its variants are well known to perform reliably on convex problems, achieving superior convergence compared to the nonconvex case. However, if a nonconvex function can be convexified through an appropriate transformation, then the guarantees of Corollary 1 can be leveraged. In this section, we analyze function classes that admit such convexifying transformations and propose several concrete examples. (The detailed proofs are deferred to the Appendix.)

We begin with repeating the simple observation: a monotonically increasing transformation $\phi$ preserves both level sets and sublevel sets.

**Claim 1.** *If the sublevel sets of $f$ are nonconvex, then $\phi \circ f$ is nonconvex for any monotone mapping $\phi$.*

While the convexity of sublevel sets is a standard property of convex functions, it is also enjoyed by pseudoconvex functions.

**Claim 2.** *If $f$ is pseudoconvex, then its sublevels sets are convex.*

Another crucial property for convexification is that the gradient should vanish only at the solution. In fact, even one-dimensional functions with vanishing gradients away from the minimizer may fail to be convexifiable.

**Example 1.** *Consider function $f(x) = |1 + (x-1)^5|$. Although its sublevel sets are convex, the function cannot be convexified via any monotone transformation. Notably, $f$ is not pseudoconvex.*

Fortunately, pseudoconvex functions avoid this pathology: their gradients do not vanish outside the solution (Lemma 1). In fact, strict pseudoconvexity is sufficient to guarantee the existence of a convexifying transformation, as we elaborate next.

## 3.1 CONVEXIFICATION

Let us analyze the Hessian of the transformed function $L = \phi \circ f$ under the monotone mapping $\phi$,

$$\nabla^2 L(x) = \phi'(f(x))\nabla^2 f(x) + \phi''(f(x))\nabla f(x)\nabla f(x)^\top.$$

Since $\phi'(f(x)) > 0$ by monotonicity, we can equivalently write

$$\nabla^2 L(x) \propto \nabla^2 f(x) + \frac{\phi''(f(x))}{\phi'(f(x))}\nabla f(x)\nabla f(x)^\top,$$

and denoting $r(x) = \frac{\phi''(f(x))}{\phi'(f(x))}$,

$$\nabla^2 L(x) \propto \nabla^2 f(x) + r(x)\nabla f(x)\nabla f(x)^\top. \tag{15}$$

Thus, convexification reduces to finding a transformation $\phi$ such that $\nabla^2 L(x)$ is positive semidefinite. For any $v \in \mathbb{R}^d$, $v^\top \nabla^2 L(x)v \propto v^\top \nabla^2 f(x)v + r(x)\left(v^\top \nabla f(x)\right)^2$. Convexity therefore requires that $v^\top \nabla^2 f(x)v > 0$ for all vectors perpendicular to the gradient, i.e., $v^T \nabla f(x) = 0$. This is precisely the first condition of pseudoconvexity stated in Lemma 1. The next step is to analyze the admissible choices of $r(x)$. To bound $r(x)$ from below, we employ the notion of the *bordered Hessian*.

**Definition 3.** *We call the bordered Hessian of twice differentiable loss $f : \mathbb{R}^d \to \mathbb{R}$ the matrix*

$$B(x) = \begin{pmatrix} 0 & \nabla f(x)^T \\ \nabla f(x) & \nabla^2 f(x) \end{pmatrix}.$$

*We denote $D_{i_1,\dots,i_k}(x)$ the principal minor of $B(x)$ of size $k + 1$, formed by rows $0, i_1, \dots, i_k$. Analogously, we denote $M_{i_1,\dots,i_k}(x)$ the principal minor of size $k$ of $\nabla^2 f(x)$, formed by rows $i_1, \dots, i_k$, with the shorthand notation $M_k(x)$ for the leading principal minor of size $k$.*

With the introduced notation we can describe the sufficient conditions on $r(x)$ for convexification.

**Theorem 2** (Schaible & Zang (1980)). *If $f$ is strictly pseudoconvex, then consider*

$$r(x) = \begin{cases} \max\{0; -1/\nabla f(x)^T\nabla^2 f(x)\nabla f(x) : M_n(x) < 0\} & \text{for strictly pseudoconvex } f, \\ \max_{i_1,\dots,i_k}\{0; M_{i_1,\dots,i_k}(x)/D_{i_1,\dots,i_k}(x) : D_{i_1,\dots,i_k} < 0\} & \text{for pseudoconvex } f \end{cases}$$

*Then, $\nabla^2 f(x) + r(x)\nabla f(x)\nabla f(x)^\top$ is positive semidefinite.*

Theorem 2 establishes the existence of a sufficient $r(x)$ to convexify any pseudoconvex function $f$ at a given point $x$. However, since these formulas depend on $x$ through $\nabla f(x)$ and $\nabla^2 f(x)$, they do not directly yield a closed-form global transformation $\phi$. For a global result, we need $r$ to depend only on functional values $f(x)$. We address this by finding a global bound $h(f(x))$ such that

$$h(f(x)) \geq r(x),$$

which can then be used to derive a valid transformation $\phi$, as stated below.

**Theorem 3.** *Let there be a global upper bound of $r(x)$ in terms of functional value, $h(f(x)) \geq r(x)$, $\forall x \in \mathbb{R}^d$. Then for any monotonically increasing mapping $\phi : [f(x_*), \infty) \to \mathbb{R}$ such that*

$$\phi(y) = \int_{f(x_*)}^{y} \exp\left(\int_{f(x_*)}^{w} h(s)ds\right) dw$$

*makes the function $\phi \circ f$ convex.*

In the theorem above, the bound is global, based on properties of $f$ over the entire unconstrained domain. However, if the algorithm satisfies monotone convergence, $f(x^{k+1}) \leq f(x^k)$ (as is typical for most second-order methods), then the iterates remain within the compact sublevel set $\mathcal{L}_{f,f(x^0)} \stackrel{\text{def}}{=} \{x \mid f(x) \leq f(x^0)\}$. In this case, properties outside the compact set are irrelevant, and the global bound $h$ can be improved by restricting it to $\mathcal{L}_{f,f(x^0)}$. For instance, the distance from the initial point to the solution, $\|x^0 - x_*\|$, can be replaced by the diameter of this compact set.

Similarly, for Newton stepsize schedules that monotonically decrease the function value, it is sufficient to consider convexification restricted to a compact set $\Delta \stackrel{\text{def}}{=} \mathcal{L}_{f,f(x^0)}$.

Table 2: Examples of radially symmetric functions of the form $f(x) = \psi(\|x - x_*\|)$ that are nonconvex, but star-convex after the transformation, see Appendix (Corollary 6, equation 25).

| Loss | Formula $\psi(x) =$ | Transformed loss $L(x) =$ | Transformation $\phi(f(x)) =$ |
|---|---|---|---|
| Geman--McClure (29) | $\frac{x^2}{x^2+1}$ | $\frac{x^2}{x^2+1} + x\arctan(x)$ | $f(x) + \sqrt{\frac{f(x)}{1-f(x)}}\arctan\left(\sqrt{\frac{f(x)}{1-f(x)}}\right)$ |
| Welsch (19) | $1 - e^{-x^2}$ | $\sqrt{\pi}x\,\mathrm{erf}(x)$ | $\sqrt{-\pi\log(1-f(x))}\,\mathrm{erf}\left(\sqrt{-\log(1-f(x))}\right)$ |
| Cauchy (9) | $\log(1+x^2)$ | $2x\arctan(x)$ | $2\sqrt{e^{f(x)}-1}\arctan\left(\sqrt{e^{f(x)}-1}\right)$ |

**Corollary 2.** *Let $f : \mathbb{R}^d \to \mathbb{R}$ be a pseudoconvex function with $r(x)$ bounded by a constant on a compact $\Delta$, $c \stackrel{\text{def}}{=} \max_{x \in \Delta} r(x) < \infty$. Then any monotonically increasing mapping $\phi : [f(x_*), \infty) \to \mathbb{R}$ such that*

$$\phi(y) = \frac{\exp\left(c(y - f(x_*))\right) - 1}{c}$$

*makes the function $\phi \circ f$ convex on the compact $\Delta$.*

For practical implementations, one can simply take $L(x) = e^{c(f(x)-f_*)}$ for sufficiently large $c$, which ensures $\nabla^2 f(x) + c\nabla f(x)\nabla f(x)^T$ is positive semidefinite. Under this exponential transformation, the stepsize multiplier is easily computed and the convergence properties from the convex setup are directly transferred to the pseudoconvex function.

### 3.2 Star-Convexification

While convexity is a powerful and widely used assumption, in some cases weaker conditions are sufficient to guarantee convergence. In particular, star-convexity is sufficient for the well-known Cubic Newton Method (Nesterov & Polyak, 2006). This motivates the construction of *star-convexifying transformations*.

To obtain convexity, one should bound the $r(x)$ by some $h(f(x))$ or constant $c$ as Theorem 3 suggests. Without closed analytical expression, this might be problematic, since the loss landscape can be intricate, and analyzing it is a separate problem. It turns out, that we can star-convexify the loss with much simpler transformation.

> **Theorem 4.** *Let $f : \mathbb{R}^d \to \mathbb{R}$ be a strictly pseudoconvex function. Then $L$ defined as below is star-convex,*
>
> $$L(x) = f(x_*) + \int_0^1 \frac{\langle \nabla f(x_* + t(x - x_*)), x - x_* \rangle}{t} dt. \qquad (16)$$

This is a more computationally friendly, as one does not have to inspect the necessary bounds and is able to calculate the function value explicitly. However, this definition requires the knowledge of the minimizer, and we previously demanded a transformation, that depended only on function value. It turns out, that if the loss function is radially symmetric, then it can be reformulated, such that it depends only on the function value.

**Corollary 3.** *For radially symmetric functions $f(x) = \psi(\|x - x_*\|)$ the transformed loss can be rewritten as following:*

$$L(x) = f(x_*) + \psi^{-1}(f(x)) \int_{f(x_*)}^{f(x)} \frac{dv}{\psi^{-1}(v)}. \qquad (17)$$

Mapping above depends only on the function value, therefore, it is a valid loss transformation. Explicit derivations for widely used robust losses can be seen in Table 2.

Since even in one-dimensional case star-convexity does not imply convexity (Nesterov & Polyak, 2006), one should carefully check the limitations of used optimizers. However, if derivatives of the initial one-dimensional function $\psi$ behave well far from the optimum, then, we might expect $g$ to be

convex after this star-convexifying transformation, so the Newton method converges.

**Lemma 3.** *If* $\psi''(r) + \frac{\psi'(r)}{r} \geq 0$ *for* $r \in [0, M]$*, then, $g$ is convex on $\mathcal{N} = \{x \mid \|x - x_*\| \leq M\}$.*

Table 3: Change of the neighborhood of the convergence for the unit-step Newton method for losses examples from Table 2. All losses are minimized at the origin.

| Loss | Formula $\psi(x) =$ | Original convergence radius | Transformed loss $L(x) =$ | Transformed convergence radius |
|---|---|---|---|---|
| Geman--McClure (29) | $\frac{x^2}{x^2+1}$ | $\frac{1}{\sqrt{3}} \approx 0.577$ | $\frac{x^2}{x^2+1} + x\arctan(x)$ | 1 |
| Welsch (19) | $1 - e^{-x^2}$ | $\frac{1}{\sqrt{2}} \approx 0.707$ | $\sqrt{\pi}x\operatorname{erf}(x)$ | 1 |
| Cauchy (9) | $\log(1 + x^2)$ | 1 | $2x\arctan(x)$ | $\infty$ |

## 4 EFFECTS OF TRANSFORMATIONS

In this section, we illustrate the impact of transformations on the behavior of Newton's method.

### 4.1 REVERSAL OF THE DESCENT DIRECTION

Our theory predicts that the scaling factor can become negative for certain losses (see Table 1), leading to a negative stepsize and hence a reversal of the descent direction. We verify this phenomenon in practice.

Figure 2 (and Figure 4 in the Appendix) confirm that such sign flips indeed occur. The regions in which the stepsize factor becomes negative depend on both the loss and the chosen transformation. For polynomial transformations with very small $r$, large portions of the domain yield negative scaling factors. For logarithmic transformations, negative scaling factors appear over large regions regardless of the parameter $a$.

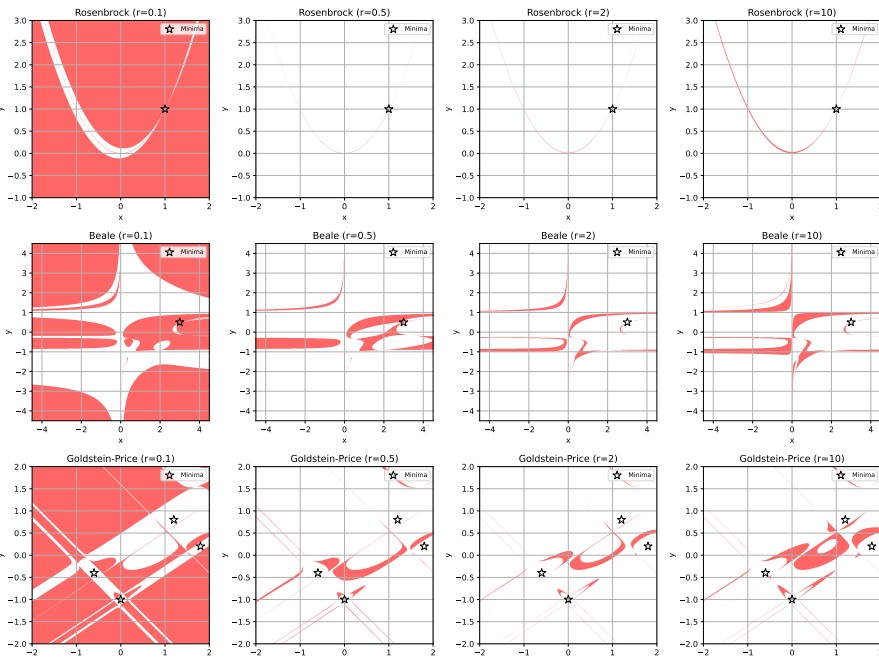

Figure 2: Red color stands for regions where the Newton step changes sign after a polynomial loss transformation.

## 4.2 CONVERGENCE NEIGHBORHOOD

Our theory also suggest that transformations can alter the convergence neighborhod of the classical Newton method. We verify this numerically: Figure 3 demonstrates how polynomial transformations change the regions of convergence.

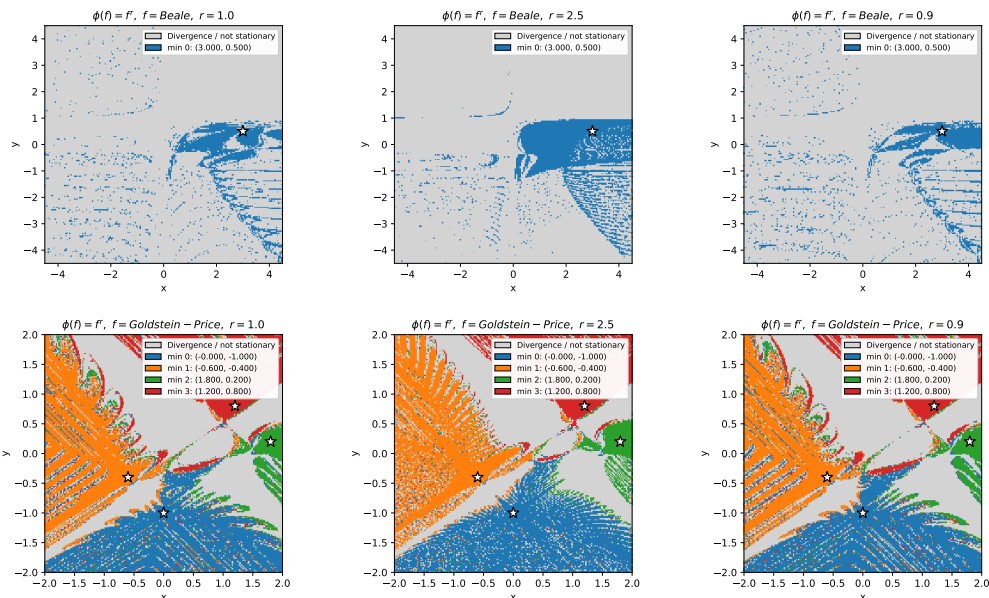

Figure 3: Convergence regions under polynomial transformations $\phi(f) = f^r$ on Beale and Goldstein–Price functions.

## 4.3 RECOVERING CONVERGENCE BY STEPSIZE RESCHEDULING

We demonstrate a practical application of transformation-induced stepsizes: recovering convergence in cases where the standard Newton method diverges.

Consider the one-dimensional function $f(x) = \ln(1 + x^2)$, which has minimizer at $x_* = 0$. If the initialization satisfies $|x_0| \geq 1/\sqrt{3}$, the unit-step Newton method diverges. For example, with $x_0 = 0.8$, the iterates diverge (left subplot of Figure 1).

If we instead transform $f$ into $L(x) = 2x \arctan(x)$ (as in Table 2) and apply the unit-step Newton method to $L$, the iterates converge to $x_*$ (middle subplot). Moreover, the sequence of iterates on $L$ can be transferred back to the original loss $f$ using the transformation-induced stepsize schedule from Corollary 1, restoring convergence (right subplot).

## 4.4 BENCHMARK LOSSES

We evaluate our approach on three benchmark objectives, each defined as a mapping $f : \mathbb{R}^2 \to \mathbb{R}$. These functions capture different challenges in optimization: the Rosenbrock loss features narrow curved valleys, the Beale loss exhibits strong nonlinearity, and the Goldstein–Price loss is multimodal. The formulas for these objectives are given by

$$f_{\text{Rosenbrock}}(x, y) = (1 - x)^2 + 100\,(y - x^2)^2, \tag{18}$$

$$f_{\text{Beale}}(x, y) = (1.5 - x + xy)^2 + (2.25 - x + xy^2)^2 + (2.625 - x + xy^3)^2, \tag{19}$$

$$f_{\text{Goldstein-Price}}(x, y) = \left[1 + (x + y + 1)^2\,(19 - 14x + 3x^2 - 14y + 6xy + 3y^2)\right]$$
$$\times \left[30 + (2x - 3y)^2\,(18 - 32x + 12x^2 + 48y - 36xy + 27y^2)\right]. \tag{20}$$

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

# A ADDITIONAL EXPERIMENTS

## A.1 SIGN FLIPS

In this section we present and visualisation of the regions with the negative scaling factors for the logarithmic loss transformation in Figure 4.

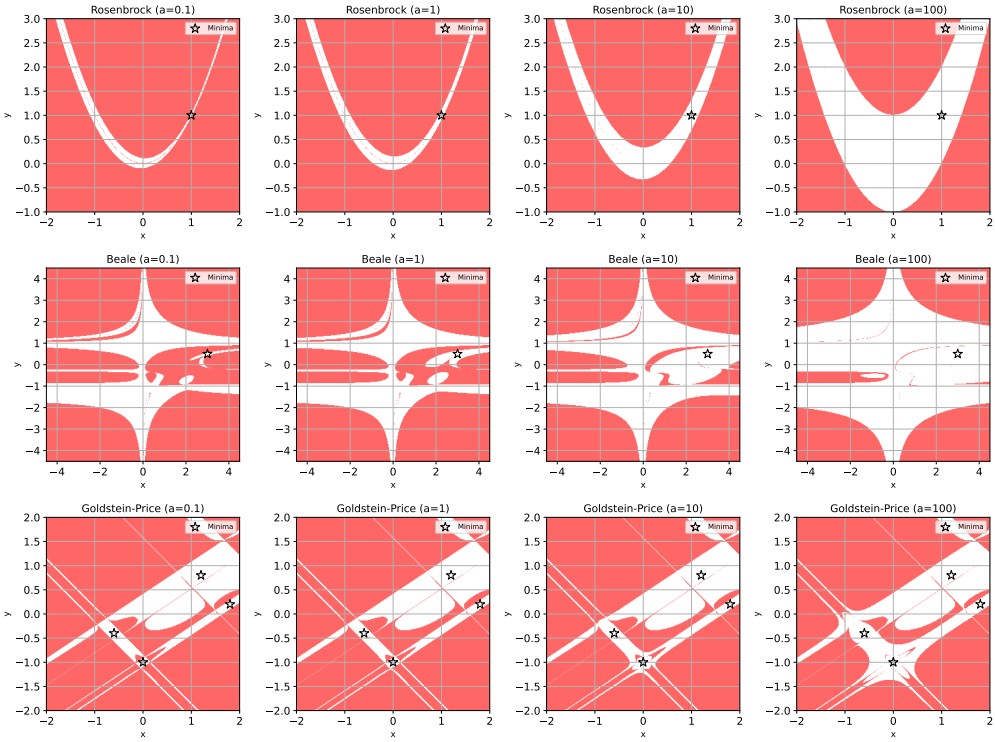

Figure 4: Red color stands for regions where the Newton step changes sign after a logarithmic loss transformation.

## A.2 2D VISUALIZATION

In this section we visualize how applying loss transformations lead to the convergence of Newton Method.

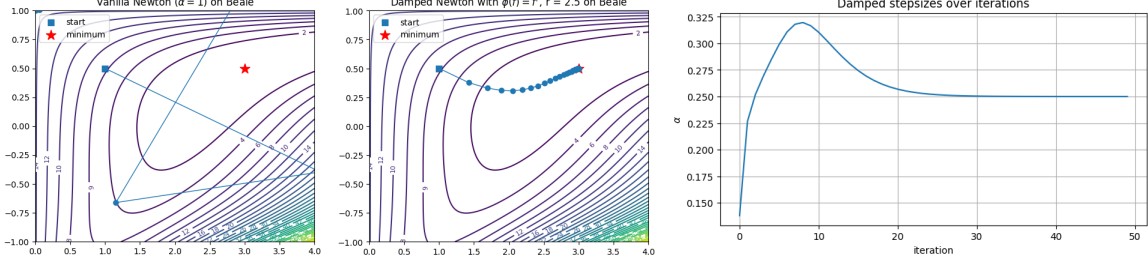

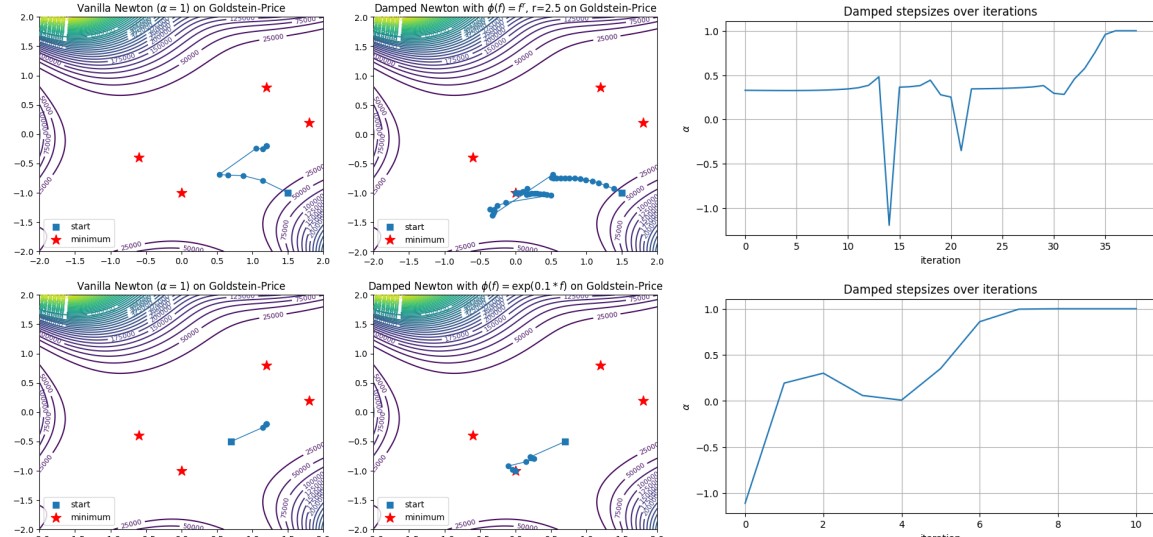

Figure 6: Visualizing trajectories of Newton method and damped Newton method after applying loss transformations. The left plot is the Vanilla Newton method. The plot in the middle is applying the loss transformations. The right plot demonstrates the change of stepsizes over the iterations. The blue square stands for the initialization point, blue circles for the iterations, and red stars for minima.

It can be seen, that applying loss transformations indeed changes the stepsizes, which benefits the convergence. These stepsizes can become either greater than one, or negative. This justifies the usage of both nonconventional stepsizes and loss transformations for second-order methods.

### A.3 Minimizing Lennard-Jones Energy

In this section we analyze the proposed transformations for real life problem, where Newton method struggles - minimization of Lennard-Jones potential for big clusters of atoms.

The Lennard-Jones (LJ) potential is a classical model used to describe pairwise interactions between neutral atoms or molecules. Given a system of $N$ particles with positions $x \in \mathbb{R}^{3N}$, the total energy is expressed as

$$E(x) = \sum_{i \neq j} 4\varepsilon \left( \left( \frac{\sigma}{r_{ij}} \right)^{12} - \left( \frac{\sigma}{r_{ij}} \right)^{6} \right),$$

where $r_{ij} = \|x_i - x_j\|$ denotes the interatomic distance, $\varepsilon$ stands for the depth of potential well and determines the attractive force between two atoms, and $\sigma$ is an effective diameter of the particle. LJ potential combines a short range repulsive term, modeling Pauli exclusion, with a long-range attractive term arising from Van der Waals forces.

Second-order methods remain one of the best numerical optimization approaches for LJ clusters (Srivastava, 2021). However, when the clusters are big ($N \geq 100$), Newton method suffers, as it requires $\mathcal{O}\left(N^3\right)$ arithmetic operations to perform a step. On the other side, Newton-MINRES are approximating $\left(\nabla^2 f(x)\right)^{\dagger} \nabla f(x)$, which is less computationally burdensome.

In Figure 7 we compare the vanilla Newton method with stepsize $\alpha = 1$ with Newton-MINRES for the cluster of $N = 300$ atoms.

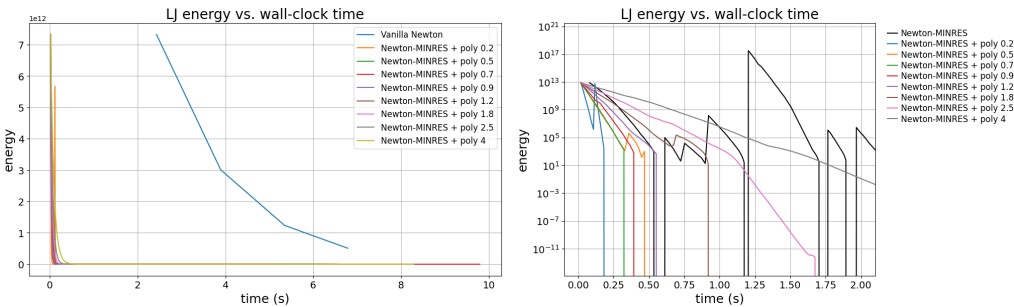

Figure 7: LJ potential minimizing with loss transformations, $N = 300$. "poly + $r$" stands for the polynomial transformation $\varphi(f) = f^r$.

Newton-MINRES with loss transformations performs significantly better, than Newton method. To decide, whether this is because of MINRES solver or the transformations themselves, in Figure 7 we also compare the Newton-MINRES with unitary stepsize. It can be seen (right plot), that applying transformations make the optimization process more stable.

In order to compare not just the trajectories, but the overall performance of the algorithm, in the Table 4 we compare the minimal LJ potential energy, obtained by Newton-MINRES methods. We compare polynomial transformations $\varphi(f) = f^r$ and logarithmic $\varphi(f) = \log(a + f)$. The latter class are significantly less robust, but the obtained minimal energy is almost the same as for the original Newton-MINRES.

| Transformation | Energy | Transformation | Energy |
|---|---|---|---|
| No transformation | -1198.58 ± 190.75 | $\varphi(f) = \log(a + f), a = 1$ | -470.21 ± 116.28 |
| $\varphi(f) = f^r, r = 0.2$ | -205.61 ± 123.41 | $\varphi(f) = \log(a + f), a = 10$ | -501.51 ± 172.14 |
| $\varphi(f) = f^r, r = 0.5$ | -275.81 ± 103.16 | $\varphi(f) = \log(a + f), a = 10^2$ | -422.00± 86.52 |
| $\varphi(f) = f^r, r = 0.7$ | -201.97 ± 166.84 | $\varphi(f) = \log(a + f), a = 10^3$ | -549.41 ± 68.76 |
| $\varphi(f) = f^r, r = 0.9$ | -219.98 ± 105.95 | $\varphi(f) = \log(a + f), a = 10^4$ | -970.88 ± 230.32 |
| $\varphi(f) = f^r, r = 1.2$ | -98.61 ± 38. 15 | $\varphi(f) = \log(a + f), a = 10^5$ | -1137 ± 213.01 |
| $\varphi(f) = f^r, r = 1.8$ | -2.31 ± 1.44 | $\varphi(f) = \log(a + f), a = 10^6$ | -1137.98 ± 213.01 |
| $\varphi(f) = f^r, r = 2.5$ | 0.00 ± 0.00 | $\varphi(f) = \log(a + f), a = 10^7$ | -1017.88 ± 207.55 |
| $\varphi(f) = f^r, r = 4$ | 0.00 ± 0.00 | $\varphi(f) = \log(a + f), a = 10^8$ | -1209.56 ± 266.16 |

Table 4: Minimal LJ potential energy for loss transformations. Mean + standard deviation across 10 runs.

## A.4 NEURAL NETWORK BENCHMARKS

In this section we reproduce the experimental setup from (Shea & Schmidt, 2024), but instead of line searches we analyze the effect of loss transformations. We take three-layer fully connected neural nets, with all the layers having hidden dimension equal to 16. As in (Shea & Schmidt, 2024) we analyze these models on benchmark LibSVM datasets (Chang & Lin, 2011). We compare the following optimizers: SGD, Adam (Kingma, 2014), SR-1 (Nocedal & Wright, 2006), L-BFGS (Nocedal & Wright, 2006), as well as SR-1 and L-BFGS with the loss transformation $\varphi(f(x)) = (f(x))^r$, as these polynomial transformations tend to perform well in practice. We took $r = 1.9$, as it has shown good robustness in training. All optimizers are ran for 50 iterates. The results can be seen in the Table 5.

Also, we report the loss curve against both iterates and wall-clock time for `a9a` dataset at Figure 8. We add SR-1 and L-BFGS with strong Wolfe condition to demonstrate, that stepsizes, inherited from loss transformations, are more effective.

| Dataset | SGD | Adam | SR-1 | L-BFGS | SR-1 + $f^r$ ($r = 1.9$) | L-BFGS + $f^r$ ($r = 1.9$) |
|---------|-----|------|------|--------|------|------|
| a1a | 0.78816 | **0.83489** | **0.83489** | 0.81308 | 0.83178 | 0.82243 |
| a9a | 0.84830 | 0.84846 | 0.85137 | 0.82589 | 0.84938 | **0.85245** |
| heart | 0.83333 | **0.87037** | 0.85185 | 0.74074 | **0.87037** | **0.87037** |
| ijcnn1 | **0.98360** | 0.94499 | 0.93499 | 0.90288 | 0.92829 | 0.97600 |
| ionosphere | 0.90141 | 0.90141 | **0.94366** | 0.90141 | 0.91549 | **0.94366** |
| splice | 0.76500 | 0.82000 | 0.83000 | 0.78000 | 0.71000 | **0.85500** |
| w1a | 0.98790 | 0.98992 | 0.97581 | 0.97177 | 0.98790 | **0.99194** |
| w8a | 0.98985 | 0.98975 | 0.97879 | 0.97095 | 0.98945 | **0.99005** |

Table 5: Accuracy for considered optimizers.

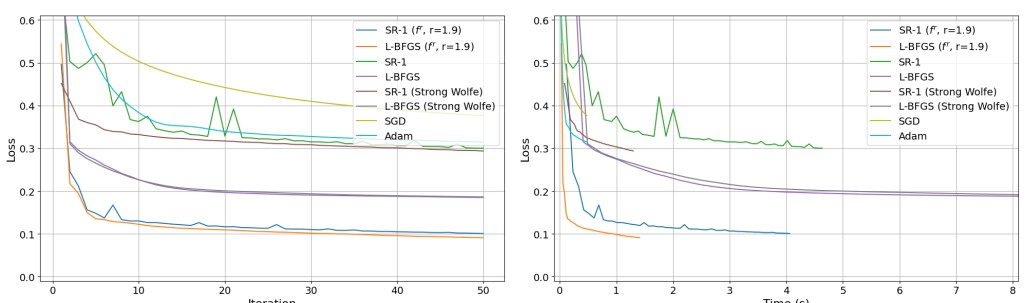

Figure 8: Loss against iteration and wall-clock time, `a9a` dataset.

## B  FURTHER CONNECTIONS TO THE LITERATURE

### B.1  NEWTON METHOD STEPSIZE RANGE

For Newton and quasi-Newton methods, virtually all explicit stepsize rules and line searches restrict stepsizes to the range $(0, 1]$. However, the optimality of this range has recently been challenged by Shea & Schmidt (2024), who show that allowing negative stepsizes can, in many cases, be more effective than restricting to positive ones only.

Our results provide a theoretical justification for this observation: under transformation invariance, the induced stepsize can naturally be larger than one or even negative. This offers a principled explanation for the effectiveness of such unconventional choices.

### B.2  NEWTON METHOD

While first-order methods often achieve satisfactory performance, incorporating second-order information can dramatically accelerate convergence. The Newton method, in particular, enjoys superlinear convergence near the optimum and performs well on convex problems, often significantly outperforming gradient descent. Nonetheless, the classical Newton method has several limitations, which have motivated numerous extensions.

Globalization techniques include adding Hessian regularization terms (Nesterov & Polyak, 2006), employing stepsize schedules (Polyak, 2009; Hanzely et al., 2022), and using Levenberg–Marquardt regularization (Levenberg, 1944; Doikov & Nesterov, 2024). When computing the exact Hessian is prohibitively expensive, algorithms with inexact Hessians are used. Quasi-Newton methods (Davidon, 1991; Fletcher, 1988) approximate the Hessian action on the gradient using heuristic updates (Nocedal & Wright, 2006) or rank constraints (Ye et al., 2021). Other approaches reuse the Hessian across multiple iterations to reduce computational cost (Doikov et al., 2023).

Preconditioning methods can also be interpreted as inexact Hessian techniques, ranging from simple diagonal approximations (Singh & Alistarh, 2020) to more sophisticated variants related to natural gradient descent (Amari, 1998). Accelerated schemes for convex problems have also been developed for convex objectives (Nesterov, 2008; Agafonov et al., 2023). Although Newton-type methods are increasingly being explored in nonconvex settings (Doikov et al., 2024b), their convergence rates and guarantees are generally weaker than in the convex case.

### B.3 Pseudoconvex Losses

Convex optimization (Tyrrell Rockafellar, 1970) has long served as the cornerstone of optimization theory, thanks to the absence of local minima and the availability of efficient algorithms. Classical methods include gradient-based techniques (Polyak, 1963), higher-order methods (Nesterov, 2021), and algorithms for constrained optimization (Frank et al., 1956). Even today, convex optimization results continue to shed light on the learning dynamics of modern machine learning models (Schaipp et al., 2025).

With the rise of deep learning, however, many practical loss landscapes are nonconvex (Cooper, 2021), motivating analysis under weaker assumptions. This has led to the development of generalized convexity frameworks (Cambini & Martein, 2009) and the study of local properties.

Quasiconvex functions (Greenberg & Pierskalla, 1971) extend convexity by requiring convex sublevel sets, enabling broader algorithmic applicability. Pseudoconvex functions (Mangasarian, 1975; Crouzeix, 2020) go further: they retain convex sublevel sets and exclude local minima, thereby preserving key advantages of convexity. Avriel & Schaible (1978) provided characterizations for differentiable and twice-differentiable functions in these classes.

Pseudoconvex losses have proven particularly useful in robust optimization (Beyer & Sendhoff, 2007; Barron, 2019). They behave like convex functions near the optimum—ensuring uniqueness of the minimizer—yet their reduced sensitivity to large deviations makes them more robust to noise. This sacrifices strict convexity while maintaining pseudoconvexity.

### B.4 Convexification

Convex optimization rates and results are often superior to nonconvex. For instance, the lower bounds for convex problems are better, than for nonconvex (Arjevani et al., 2015; 2023). Furthermore, results for the function value can be obtained (Nesterov, 2018), as well as the last-iterate bounds (Defazio et al., 2024). Therefore, whether convex reformulation of the problem exist, it will drastically improve the obtained result. This was analyzed thoroughly in mathematical programming, where changing the minimized functional, as well as the constraints and the feasible set, might lead to the same problem, but with the convexity (Charnes & Cooper, 1962; Fujie & Kojima, 1997; Günlük & Linderoth, 2011; Xia, 2020).

Another link between convex problems and nonconvex is the concept of hidden convexity (Fatkhullin et al., 2025), which claims, that nonconvex problem $\min_{x \in \mathbb{R}^d} F(x)$ might be rewritten as $\min_{u \in \mathbb{R}^d} H(c^{-1}(u))$, where $H$ is convex. This is present in optimal control (Anderson et al., 2019), reinforcement learning Zhang et al. (2020), and revenue management (Chen et al., 2025). While analyzed mostly for first-order methods, there were attempts to incorporate the Newton method with this concept, for instance, in (Izmailov & Solodov, 2025) they analyzed the inexactness between the exact solution and the transformed one. The key point in these analyses is the transformation of the variables, which is done by mapping $c : \mathbb{R}^d \to \mathbb{R}^d$. We, on the other hand, consider one-dimensional mapping $\phi : \mathbb{R} \to \mathbb{R}$. This makes the access to the value and its derivatives significantly easier, as we avoid storing and multiplying large matrices and tensors.

## C Preservation of Newton Method's Iterations

### C.1 Proof of Theorem 1

*Proof of Theorem 1.* **Notation.** Throughout this proof let

$$g \stackrel{\text{def}}{=} \nabla f(x), \quad \mathbf{H} \stackrel{\text{def}}{=} \nabla^2 f(x), \quad \phi' \stackrel{\text{def}}{=} \phi'(f(x)), \quad \phi'' \stackrel{\text{def}}{=} \phi''(f(x)).$$

For the transformed loss $L(x) = \phi(f(x))$ we have

$$\nabla L(x) = \phi' g, \qquad \nabla^2 L(x) = \phi'' g g^\top + \phi' \mathbf{H}.$$

Abbreviate $x = x_k$ and $x_{k+1} = x^+$.

Starting from the transformed step and substituting $\nabla L, \nabla^2 L$,

$$x^+ - x = -\alpha_L(x)\left(\nabla^2 L(x)\right)^\dagger \nabla L(x)$$

$$= -\alpha_L\left(\phi'' \, gg^\top + \phi' \, \mathbf{H}\right)^\dagger \phi' \, g.$$

Now we need to carefully work with the pseudoinverse. Define $\mathbf{A} = \phi'' gg^T + \phi' \mathbf{H}$. First of all, we show $\mathbf{A}^\dagger g \in \mathrm{Range}(\mathbf{H})$. Since $\mathbf{H}$ is symmetric, $\mathbb{R}^d = \mathrm{Range}(\mathbf{H}) \oplus \ker(\mathbf{H})$. Therefore, for any vector $v = v_R + v_0$, where $v_R \in \mathrm{Range}(H)$, $v_0 \in \ker(\mathbf{H})$ we have

$$\mathbf{A}v = \phi'\mathbf{H}v_R + \phi'\mathbf{H}v_0 + \phi'' gg^T v_R + \phi'' gg^T v_0 = \phi'\mathbf{H}v_R + \phi'' gg^T v_R = \mathbf{A}v_R,$$

as $g \in \mathrm{Range}(\mathbf{H})$ and $\mathrm{Range}(\mathbf{H}) \perp \ker(\mathbf{H})$. Therefore, we have $\mathrm{Range}(\mathbf{A}) \subset \mathrm{span}\left(\mathrm{Range}(\mathbf{H}) \cup \mathrm{span}(g)\right) = \mathrm{Range}(\mathbf{H})$, and

$$\mathbf{A}^\dagger g = \underset{v\in\mathbb{R}^d}{\mathrm{argmin}}\left\{\|v\| \mid v \in \underset{y\in\mathbb{R}^d}{\mathrm{argmin}}\|\mathbf{A}y - g\|\right\} = \underset{v\in\mathrm{Range}(\mathbf{H})}{\mathrm{argmin}}\left\{\|v\| \mid v \in \underset{y\in\mathrm{Range}(\mathbf{H})}{\mathrm{argmin}}\|\mathbf{A}y - g\|\right\}.$$

Hence, we obtained $\mathbf{A}^\dagger g \in \mathrm{Range}(\mathbf{H})$.

Next, we examine the action of $\mathbf{A}$. Since $g \in \mathrm{Range}(\mathbf{H})$, there exist $u$, such that $\mathbf{H}u = g$. Among all vectors, take $u$ with the minimal norm, i.e. $u = \mathbf{H}^\dagger g$. Therefore, for any $y \in \mathbb{R}^d$ we have

$$\mathbf{A}y = \phi'\mathbf{H}y + \phi'' gg^T y = \phi'\mathbf{H}y + \phi''(\mathbf{H}u)g^T y = \mathbf{H}(\phi'\mathbf{I} + \phi'' ug^T)y.$$

Hence,

$$\mathbf{A} = \mathbf{H}(\phi'\mathbf{I} + \phi'' ug^T). \tag{21}$$

We aim to find the $v = \mathbf{A}^\dagger g$. We have $\mathbf{A}v = \mathbf{A}\mathbf{A}^\dagger g = \mathrm{Proj}_{\mathrm{Range}(\mathbf{A})}\, g$. Therefore, if $g \in \mathrm{Range}(\mathbf{A})$, we obtain $\mathbf{A}v = g$. To ensure this, we utilize the fact $\phi' + \phi''\|g\|_x^{*2} \neq 0$:

$$\mathbf{A}u = (\phi'\mathbf{H} + \phi'' gg^T)u = \phi'\mathbf{H}u + \phi'' g(g^T u) = \phi' g + \phi''(g^T \mathbf{H}^\dagger g)g = (\phi' + \phi''\|g\|_x^{*2})g.$$

Consequently, we have $\mathbf{A}v = g$, and since $v \in \mathrm{Range}(\mathbf{H})$, from equation 21 we obtain

$$\mathbf{A}v = \mathbf{H}(\phi'\mathbf{I} + \phi'' ug^T)v = g = \mathbf{H}u.$$

Apply $\mathbf{H}^\dagger$ to both parts of the equation:

$$\mathrm{Proj}_{\mathrm{Range}(\mathbf{H})}((\phi'\mathbf{I} + \phi'' ug^T)v) = \mathbf{H}^\dagger\mathbf{H}(\phi'\mathbf{I} + \phi'' ug^T)v = \mathbf{H}^\dagger\mathbf{H}u = \mathrm{Proj}_{\mathrm{Range}(\mathbf{H})}(u).$$

Since $\mathbf{H}$ is symmetric, we have $\mathrm{Range}(\mathbf{H}) = \mathrm{Range}(\mathbf{H}^\dagger)$. Therefore, $u = \mathbf{H}^\dagger g \in \mathrm{Range}(\mathbf{H})$. Since $v = \mathbf{A}^\dagger g$, $\mathbf{A} = \phi'\mathbf{H} + \phi'' gg^T$ is symmetric, and $\mathrm{Range}(\mathbf{A}) \subset \mathrm{Range}(\mathbf{H})$, we also result in $v \in \mathrm{Range}(\mathbf{A}) \subset \mathrm{Range}(\mathbf{H})$. And $(\phi'\mathbf{I} + \phi'' ug^T)v = \phi'v + \phi''(g^T v)u \in \mathrm{Range}(\mathbf{H})$. Hence,

$$(\phi'\mathbf{I} + \phi'' ug^T)v = u.$$

According to Sherman-Morrison formula if and only if $\phi' \neq 0$, $1 + \frac{\phi''}{\phi'}g^T u \neq 0$, on the whole $\mathbb{R}^d$ we have

$$(\phi'\mathbf{I} + \phi'' ug^T)^{-1} = \frac{1}{\phi'}\left(\mathbf{I} - \frac{\frac{\phi''}{\phi'}}{1 + \frac{\phi''}{\phi'}g^T u}ug^T\right) = \frac{1}{\phi'}\left(\mathbf{I} - \frac{\phi''}{\phi' + \phi''\|g\|_x^{*2}}ug^T\right).$$

The condition $\phi' \neq 0$ is satisfied by assumption, and $1 + \frac{\phi''}{\phi'}g^T u = \frac{\phi' + \phi''\|g\|_x^{*2}}{\phi'}$. Therefore, with $\phi' + \phi''\|g\|_x^{*2} \neq 0$ it is satisfied. Hence, applying this inverse to $u$ we have

$$v = (\phi'\mathbf{I} + \phi'' ug^T)^{-1}u = \frac{1}{\phi'}\left(\mathbf{I} - \frac{\phi''}{\phi' + \phi''\|g\|_x^{*2}}ug^T\right)u$$

$$= \frac{1}{\phi'}\left(u - \frac{\phi''}{\phi' + \phi''\|g\|_x^{*2}}(g^T u)u\right) = \frac{1}{\phi'}\left(\frac{\phi' + \phi''\|g\|_x^{*2} - \phi'' g^T u}{\phi' + \phi''\|g\|_x^{*2}}\right)u$$

$$= \frac{1}{\phi'}\frac{\phi'}{\phi' + \phi''\|g\|_x^{*2}}u = \frac{1}{\phi' + \phi''\|g\|_x^{*2}}u.$$

Collecting everything altogether we obtain

$$x^+ - x = -\alpha_L\mathbf{A}^\dagger\phi' g == -\alpha_L\frac{\phi'}{\phi' + \phi''\|g\|_x^{*2}}\mathbf{H}^\dagger g = -\alpha\,\mathbf{H}^\dagger g,$$

with the scalar stepsize

$$\alpha = \alpha_L \frac{\phi'}{\phi' + \phi'' \|g\|_x^{*2}}.$$

This is exactly the claimed form. □

**Corollary 4.** *If* $\mathbf{H}$ *is invertible, then,* $g \in \text{Range}(\mathbf{H}) = \mathbb{R}^d$.

## C.2 PROOF OF LEMMA 2

It can be shown, that even with invertible matrices, Levenberg-Marquardt regularization cannot be considered transformation invariant.

*Proof of Lemma 2.* Assume, for contradiction, that an LM regularization on $f$ is transformation-invariant. Let

$$g \stackrel{\text{def}}{=} \nabla f(x), \quad \mathbf{H} \stackrel{\text{def}}{=} \nabla^2 f(x), \quad \phi'_x \stackrel{\text{def}}{=} \phi'(f(x)), \quad \phi''_x \stackrel{\text{def}}{=} \phi''(f(x)), \quad \lambda_\phi \stackrel{\text{def}}{=} \lambda_\phi(x, \phi(\cdot), \lambda(x)).$$

Suppose the (damped) Newton step satisfies

$$-(x^+ - x) = (\mathbf{H} + \lambda(x)\mathbf{I})^{-1} g = (\nabla^2 L(x) + \lambda_\phi \mathbf{I})^{-1} \nabla L(x)$$

$$= (\phi''_x gg^\top + \phi'_x \mathbf{H} + \lambda_\phi \mathbf{I})^{-1} \phi'_x g.$$

Equivalently,

$$(\mathbf{H} + \lambda(x)\mathbf{I})^{-1} g = \left(\mathbf{H} + \frac{\phi''_x}{\phi'_x} gg^\top + \frac{\lambda_\phi}{\phi'_x} \mathbf{I}\right)^{-1} g.$$

For generic $(\mathbf{H}, g)$ this identity imposes $d$ scalar equations on the single unknown $\lambda_\phi$ (the rank-one term depends on $g$), so it admits no solution in general. Thus a scalar LM coefficient cannot be chosen to make the regularized step invariant under $\phi \circ f$. □

# D EXAMPLES OF NON-CONVEXIFIABLE LOSSES

**Example 2** (Nonconvex sublevel sets). *Let* $\mathcal{L}_{f,c} := \{x : f(x) \le c\}$. *Assume there exist* $x, y \in \mathcal{L}_{f,c}$ *and* $t \in (0,1)$ *such that* $z := tx + (1-t)y \notin \mathcal{L}_{f,c}$, *i.e.,*

$$f(z) > c \ge \max\{f(x), f(y)\}.$$

*For any monotone* $\phi$, *we have* $x, y \in \mathcal{L}_{\phi \circ f, \max\{\phi(f(x)), \phi(f(y))\}}$, *while by monotonicity,*

$$z \notin \mathcal{L}_{\phi \circ f, \max\{\phi(f(x)), \phi(f(y))\}}.$$

*Hence* $\phi \circ f$ *has a nonconvex sublevel set and therefore cannot be convex.*

**Example 3** (Nonconvexifiability of $f(x) = |1 + (x-1)^5|$). *Let* $x_1 = 1$ *and* $x_0 := 1 - 2^{1/5}$. *Then* $f(x_1) = f(x_0) = 1$, *but* $f'(x_1) = 0$ *while* $f'(x_0) \ne 0$. *Suppose there exists a monotone* $\phi$ *with* $\phi(0) = 0$, $\phi(1) = 1$ *such that* $L = \phi \circ f$ *is convex. For any* $\varepsilon > 0$, *since* $1 = \frac{\varepsilon \cdot 0 + 1 \cdot (1+\varepsilon)}{1+\varepsilon}$, *convexity gives*

$$L(1) \le \frac{\varepsilon}{1+\varepsilon} L(0) + \frac{1}{1+\varepsilon} L(1+\varepsilon) \implies 1 \le \frac{L(1+\varepsilon) - L(1)}{\varepsilon}.$$

*Now* $L(1+\varepsilon) = \phi(1+\varepsilon^5)$, *hence*

$$\frac{1}{\varepsilon^4} \le \frac{\phi(1+\varepsilon^5) - \phi(1)}{\varepsilon^5}.$$

*Letting* $\varepsilon \downarrow 0$, *we obtain the right derivative* $\phi'_+(1) = +\infty$. *By the chain rule where* $f$ *is differentiable,* $L'(x) = \phi'(f(x)) f'(x)$, *so at* $x_0$ *we would have* $|L'(x_0)| = +\infty$ (*since* $f(x_0) = 1$ *and* $f'(x_0) \ne 0$), *which is impossible for a finite-valued convex function on* $\mathbb{R}$. *Therefore no monotone* $\phi$ *can convexify* $f$.

# E MISSING PROOFS FOR (STAR-)CONVEXIFICATION

**Theorem 5** (Theorem 3 from the main part). *Let there be a global upper bound of* $r(x)$ *in terms of functional value,* $h(f(x)) \ge r(x)$, $\forall x \in \mathbb{R}^d$. *Then for any monotonically increasing*

*mapping $\phi : [f(x_*), \infty) \to \mathbb{R}$ such that*

$$\phi(y) \geq \int_{f(x_*)}^{y} \exp\left(\int_{f(x_*)}^{w} h(s)ds\right) dw$$

*makes the function $\phi \circ f$ convex.*

*Proof.* Theorem 2 provides a function $r(x)$, such that $\nabla^2 f(x) + r(x)\nabla f(x)\nabla f(x)^T$ is positive semidefinite. Since $h(f(x)) \geq r(x)$, then, $\forall y \in \mathbb{R}^d$ we have

$$y^T \left(\nabla^2 f(x) + h(f(x))\nabla f(x)\nabla(x)^T\right) y =$$
$$y^T \left(\nabla^2 f(x) + r(x)\nabla f(x)\nabla(x)^T\right) y + (h(f(x)) - r(x)) y^T \nabla f(x)\nabla f(x) y \geq$$
$$(h(f(x)) - r(x)) y^T \nabla f(x)\nabla f(x) y = (h(f(x)) - r(x)) (y^T \nabla f(x))^2 \geq 0, \qquad (22)$$

hence, $\nabla^2 f(x) + h(f(x))\nabla f(x)\nabla f(x)^T$ is also positive semidefinite.

To satisfy the convexity of the transformed loss $L(x) = \phi(f(x))$, we need its Hessian $\nabla^2 L(x) = \phi'(f(x))\nabla^2 f(x) + \phi''(f(x))\nabla f(x)\nabla f(x)^T$ to be positive semidefinite. Since $\phi$ is strictly increasing, it is sufficient and necessary to show the positive semidefiniteness of $\nabla^2 f(x) + \frac{\phi''(f(x))}{\phi'(f(x))}\nabla f(x)\nabla f(x)^T$. Similarly to 22, it can be shown, that if

$$\frac{\phi''(f(x))}{\phi'(f(x))} \geq h(f(x)), \qquad (23)$$

then, $\nabla^2 f(x) + \frac{\phi''(f(x))}{\phi'(f(x))}\nabla f(x)\nabla f(x)^T$ is also positive semidefinite.

Define $g(y) = \phi'(y), y_* = f(x_*)$ W.l.o.g. assume $\phi(y_*) = 0, \phi'(y_*) = 1$. Then, to obtain eq. (23) we need following:

$$g'(y) \geq g(y)h(y)$$
$$g(y) \geq g(y_*)\exp\left(\int_{y_*}^{y} h(s)ds\right)$$
$$\phi'(y) \geq \exp\left(\int_{y_*}^{y} h(s)ds\right)$$
$$\phi(y) = \int_{y_*}^{y} \exp\left(\int_{y_*}^{w} h(s)ds\right) dw.$$

$\square$

**Corollary 5** (Corollary 2 from the main part). *Let $f : \mathbb{R}^d \to \mathbb{R}$ be a pseudoconvex function with $r(x)$ bounded by a constant on a compact $\Delta$, $c \overset{def}{=} \max_{x \in \Delta} r(x) < \infty$. Then any monotonically increasing mapping $\phi : [f(x_*), \infty) \to \mathbb{R}$ such that*

$$\phi(y) = \frac{\exp(c(y - f(x_*))) - 1}{c}$$

*makes the function $\phi \circ f$ convex on the compact $\Delta$.*

*Proof.* We can define $h(f(x)) = c$. Therefore,

$$\phi(y) \geq \exp\left(\int_{y_*}^{y} \exp\left(\int_{y_*}^{w} cds\right) dw\right) = \exp\left(\int_{y_*}^{y} \exp(c(w - y_*))dw\right) =$$
$$= \frac{1}{c}\left(e^{c(y - y_*)} - 1\right)$$

is sufficient for convexification on $\Delta$ $\square$

**Theorem 6** (Theorem 4 from the main part). *Let $f : \mathbb{R}^d \to \mathbb{R}$ be a strictly pseudoconvex function. Then $L$ defined as below is star-convex,*

$$L(x) = f(x_*) + \int_0^1 \frac{\langle \nabla f(x_* + t(x - x_*)), x - x_* \rangle}{t} dt. \tag{24}$$

*Proof.* First of all, we show, that this definition is correct, i.e. $L$ exists. For twice differentiable $f$ we have

$$\nabla f(x + u) = \nabla f(x) + \nabla^2 f(x)u + o(u), \quad u \to 0.$$

Therefore,

$$\frac{\langle \nabla f(x_* + t(x - x_*))x - x_*, \rangle}{t} = \frac{t \langle \nabla^2 f(x_*)(x - x_*), x - x_* \rangle + o(t)}{t}$$

$$= \langle \nabla^2 f(x_*)(x - x_*), x - x_* \rangle + o(1), \quad t \to 0.$$

Hence, integral converges. Then, we shall prove the star-convexity of $L$. For $\lambda \in [0, 1]$ we have

$$L((1 - \lambda)x_* + \lambda x) = L(x_* + \lambda(x - x_*))$$

$$= f(x_*) + \int_0^1 \frac{\langle \nabla f(x_* + t\lambda(x - x_*)), \lambda(x - x_*) \rangle}{t} dt$$

$$= f(x_*) + \lambda \int_0^\lambda \frac{\langle \nabla f(x_* + s(x - x_*)), (x - x_*) \rangle}{s} ds$$

As $f(x_* + s(x - x_*)) > f(x_*)$, we have

$$0 < \langle \nabla f(x_* + s(x - x_*)), s(x_* - x) \rangle = s \langle \nabla f(x_* + s(x - x_*)), x_* - x \rangle.$$

Hence,

$$L(x_* + (1 - \lambda)x) = f(x_*) + \lambda \int_0^\lambda \frac{\langle \nabla f(x_* + s(x - x_*)), (x - x_*) \rangle}{s} ds$$

$$\leq f(x_*) + \lambda \int_0^1 \frac{\langle \nabla f(x_* + s(x - x_*)), (x - x_*) \rangle}{s} ds$$

$$= (1 - \lambda)L(x_*) + \lambda L(x).$$

$\square$

**Corollary 6** (Corollary 3 from the main part). *For radially symmetric functions $f(x) = \psi(\|x - x_*\|)$ the transformed loss can be rewritten as following:*

$$L(x) = f(x_*) + \|x - x_*\| \int_0^{\|x - x_*\|} \frac{\psi'(t)}{t} dt = f(x_*) + \psi^{-1}(f(x)) \int_{f(x_*)}^{f(x)} \frac{dv}{\psi^{-1}(v)} \tag{25}$$

*Proof.* If $f(x) = \psi(\|x - x_*\|)$, then, $\nabla f(x) = \psi'(\|x - x_*\|) \frac{x - x_*}{\|x - x_*\|}$. Therefore, $L$ can be rewritten as following:

$$L(x) = f(x_*) + \int_0^1 \frac{\|x - x_*\| \psi'(t\|x - x_*\|)}{t} dt = f(x_*) + \|x - x_*\| \int_0^{\|x - x_*\|} \frac{\psi'(s)}{s} ds.$$

If $v = \psi(s)$, then, $dv = \psi'(s)ds$. Therefore, $\frac{\psi'(s)}{s} ds = \frac{dv}{s} = \frac{dv}{\psi^{-1}(v)}$, and we can rewrite

$$L(x) = f(x_*) + \psi^{-1}(f(x)) \int_{f(x_*)}^{f(x)} \frac{dv}{\psi^{-1}(v)}.$$

$\square$

**Lemma 4** (Lemma 3). *If $\psi''(r) + \frac{\psi'(r)}{r} \geq 0$ for $r \in [0, M]$, then, $L$ is convex on $\mathcal{N} = \{x \mid \|x - x_*\| \leq M\}$.*

*Proof.* Let $r = \|x - x_*\|$ and $u = \frac{x - x_*}{r}$. Define $\Psi(r) = r \int_0^r \frac{\psi'(t)}{t} dt$, so $L(x) = f(x_*) + \Psi(r)$.

By the product rule we get:

$$\Psi'(r) = \int_0^r \frac{\psi'(t)}{t} dt + \psi'(r), \quad \Psi''(r) = \frac{\psi'(r)}{r} + \psi''(r).$$

Then, we obtain

$$\nabla L(x) = \Psi'(r)u$$

and

$$\nabla^2 L(x) = \Psi''(r)uu^T + \frac{\Psi'(r)}{r}\left(\mathbf{I} - uu^T\right).$$

Therefore, there are 2 eigenvalues: $\lambda_1 = \Psi''(r)$ and $\lambda_2 = \frac{\Psi'(r)}{r}$, which we need to examine.

$\lambda_1 = \Psi''(r) = \psi''(r) + \frac{\psi'(r)}{r} \geq 0$, as we assumed.

Since $\nabla f(x_*) = 0$, we have $\psi'(0) = 0$. Using this fact, we obtain $\lambda_2 = \frac{\Psi'(r)}{r} = \frac{1}{r}\left(\int_0^r \frac{\psi'(t)}{t} dt + \psi'(r)\right) = \frac{1}{r}\int_0^r \left(\frac{\psi'(t)}{t} + \psi''(t)\right) dt \geq 0$.

This is valid for $r > 0$. Since $\psi \in C^2([0, M])$ as is f, and $\lim_{r \to 0} \frac{\psi'(r)}{r} = \psi''(0)$, these eigenvalues are continuous. Therefore, for $r = 0$ these eigenvalues are also nonnegative. $\square$

For losses from Table 2, we can derive first and second derivatives,

$$\phi_1'(c) = 1 + \frac{1}{2\sqrt{c(1-c)^3}} \int_0^c \sqrt{\frac{1-v}{v}} \, dv$$

$$\phi_1''(c) = \frac{-1 + 2c}{4\left(c(1-c)^3\right)^{3/2}} \int_0^c \sqrt{\frac{1-v}{v}} \, dv + \frac{1}{2\sqrt{c(1-c)^3}}$$

$$\phi_2'(c) = 1 + \frac{e^c}{2\sqrt{e^c - 1}} \int_0^c \frac{dv}{\sqrt{e^v - 1}}$$

$$\phi_2''(c) = \frac{e^c(e^c - 2)}{4(e^c - 1)^{3/2}} \int_0^c \frac{dv}{\sqrt{e^v - 1}} + \frac{e^c}{2\sqrt{e^c - 1}}$$

$$\phi_3'(c) = 1 + \frac{1}{2(1-c)\sqrt{-\log(1-c)}} \int_0^c \frac{dv}{\sqrt{-\log(1-v)}}$$

$$\phi_3''(c) = \frac{2 + \log(1-c)}{4(1-c)^2(-\log(1-c))^{3/2}} \int_0^c \frac{dv}{\sqrt{-\log(1-v)}} + \frac{1}{2(1-c)\sqrt{-\log(1-c)}}$$

