# OpenReview forum: "Loss Transformation Invariance of the Damped Newton Methods"
_ICLR.cc/2026/Conference — Submitted to ICLR 2026_

### Official Review · Reviewer_dgoH · 2025-10-27

**Soundness:** 1
**Presentation:** 4
**Contribution:** 2
**Rating:** 4
**Confidence:** 3

**Summary:**

This paper investigates loss transformation invariance in the damped Newton method, showing that the Newton stepsize schedules are invariant under monotonic loss transformations.
The authors prove that, by carefully designing such transformations, applying the damped Newton method to the original objective function $f(x)$ is equivalent to applying the vanilla Newton method to a transformed objective function $L(x)$, extending its applicability to certain nonconvex problems.
The paper also offers insights for why stepsizes greater than one and negative stepsizes can still be effective in second-order methods.

**Strengths:**

1. The main theory results are innovative, which provide explanations for several empirical observations such as why stepsizes larger than one and negative stepsizes can be effective for second-order methods.

2. The motivations for using loss transformation and stepsize scheduling are clear, and the writing flows smoothly, from the discussion on convex and pseudoconvex problems to the convexification of a broad class of nonconvex problems.

**Weaknesses:**

1. Such loss transformation assumes the objective function is pseudoconvex (sufficient) or its sublevel sets are convex (necessary), which are quite strong for real-world problems and difficult to verify.

2. All experiments are conducted on 1D and 2D synthetic problems which are insufficient. Larger and higher-dimensional problems should be included. Moreover, since Theorems 3 and 4 are stated for $x \in \mathbb{R}^d$, it is necessary to include higher-dimensional experiments to validate these claims.

3. Only the vanilla Newton method and the damped Newton method are compared. Other second-order baselines, such as quasi-Newton, cubic regularization, trust region, and Newton-Krylov methods, should also be evaluated on large, real-world nonconvex problems. It would be helpful to show whether the damped Newton method achieves better robustness, or whether the stepsize scheduling achieves better efficiency (in time) than traditional strategies like line search.

4. Some minor typos:
    - lines 41 - 42: should be $O(1/k^2)$ instead of $O(1/k^-2)$, and similar elsewhere.
    - line 62: "than" -> "then"
    - line 274: clarify the phrasing "strict pseudoconvexity is sufficient properties are sufficient"
    - line 278: should be $\phi \circ f$ instead of $f \circ \phi$

**Questions:**

1. Theorem 3 assumes a global upper bound. How can such a bound be defined, or what are the consequences if it is not tight?

---

> ### Author Response · Authors · 2025-11-24
>
> We thank the reviewer for their insightful and constructive remarks and are ready to answer the raised concerns.
>
> 1)Is pseudoconvexity a strict assumption in real life?
>
> As mentioned in the related literature, pseudoconvex losses are used in the robust optimization, where models need to be less influenced by outliers [1]. Furthermore, theory for pseudoconvex optimization might be relevant to large scale problems, as it was shown with the convex optimization [2]. As we had shown in the experiment section, loss transformations indeed enlarge the area of convergence of the damped Newton method.
>
> 2)Larger and higher-dimensional problems should be included
>
> We have provided 1D and 2D problems to visualize the increasing area of convergence. Since Theorems 3 and 4 describe the convexification, we are not clear what high-dimensional experiments are required to illustrate them properly. We will be thankful, if you specify them.
>
> 3What about other baselines: Cubic Newton, Quasi-Newton, Trust Region, Newton-Krylov?
>
> The key point of our paper is not to compare the performance of methods, but to investigate the invariance of the damped Newton method under the loss transformations. Analyzing the invariance of the mentioned methods we obtain:
>
> **Cubic Newton**
>
> The vanilla cubic Newton falls into the type of Levenberg-Marquardt regularization, as the solution
>
> $$ x^{k+1} = \operatorname{argmin}_{y\in \mathbb{R}^d} \\{ f(x^k) + \langle \nabla f(x^k),y-x> + \frac{1}{2}\langle \nabla^2f(x^k)(y-x^k),y-x^k\rangle + \frac{M}{6}\|y-x^k\|^3 \\}$$
>
> can be rewritten, as
>
> $$x^{k+1} = x^k - \left(\nabla^2 f(x^k) + \frac{M}{2}\|x^{k+1}-x^k\| \cdot I\right)^{-1}\nabla f(x^k)  =x^k - \left(\nabla^2 f(x^k) +\lambda \cdot I\right)^{-1}\nabla f(x^k).$$
>
> Lemma 2 tells ,that for general ($\nabla^2 f(x^k), \nabla f(x^k)$) there might be no loss coefficient, that preserves loss invariance.
>
> **Quasi-Newton**
>
> Quasi-Newton methods are based on a secant equation. Below we show that this equation does not allow the loss transformation invariance.
>
> If we demand the same iterations of the damped Quasi-Newton method, we obtain
>
> $$ -\alpha_f B_f^{-1} \nabla f(x) =  x^+ - x = - \alpha_L H_L \nabla L(x) =  - \alpha_L \varphi’(f(x)) B_L^{-1} \nabla f(x).$$
>
> Therefore, we require simultaneously
>
> $$ B_f (x^+ - x) = \alpha_f \nabla f(x),~~~ B_L(x^+ - x) = \alpha_L \varphi’(f(x)) \nabla f(x),$$
>
> so these vectors in the right side are collinear. But the secant equation results in
>
> $$B_f (x^+ - x) = \nabla f(x^+) - \nabla f(x), ~~~ B_L(x^+ - x) =\nabla L(x^+) - \nabla L(x) = \varphi’(f(x^+))\nabla f(x^+) - \varphi’(f(x))\nabla f(x).$$
>
> While the left side is the same as previously, vectors in the right side are no longer collinear, unless $\phi$ is affine.
>
> **Trust Region**
>
> Trust region heavily affects the direction of the update, since the minimum can be attained at the boundary. This depends on the approximation and the trust region radius simultaneously, therefore, there is no uniform theory, that can be analyzed, like for the damped Newton method in the manuscript
>
> **Newton-Krylov**
>
> Newton-Krylov method is in fact the Newton Method, but with Krylov subspace-based subsolver to compute the required matrix-vector product. Hence, this does not change the iteration of the damped Newton method, it is also loss invariant.
>
>
> 4)What are the consequences of not tight bound in Theorem 3 (Theorem 5 in appendix)?
>
> Indeed, the inequality in Theorem 3 might lead to different convergence rates, depending on its looseness. W.l.o.g. the key point is in analyzing the properties of the function $\phi(x) = e^{c( f(x) - f_*)}$ (Corollary 2), such as  Lipschitz constant, strongly convexity constant, and self-concordance constant. Therefore, working on a compact, these constants scale exponentially. Hence, the convergence will be slower with large $c$. However, we should additionally mention that without this mapping $\phi$, Newton method will diverge for any initialization.
>
> Consequently, loose inequality in Theorem 3 might lead to poorer convergence in comparison to less loose inequality. But it depends heavily on the initial point, the choice of function $h$ and function itself. Nevertheless, with any of these “bad choices” of $h$, the method will converge, which is not observed for vanilla Newton method.
>
> Also, Theorem 1 provides a transfer of stepsizes, therefore, any damped Newton stepsize scheduler for convex problems can be incorporated. It will obtain the same convergence results for pseudoconvex ones.
>
> 6)Typos
>
> Thanks, we fixed them!
>
> [1] Barron, Jonathan T. "A general and adaptive robust loss function."
>
> [2]Schaipp, Fabian, et al. "The Surprising Agreement Between Convex Optimization Theory and Learning-Rate Scheduling for Large Model Training."
>
> [3] Horst, Reiner. "On the convexification of nonlinear programming problems: An applications-oriented survey."
>
> [4] Schaible, Siegfried, and Israel Zang. "On the convexifiability of pseudoconvex C2-Functions."

---

> > ### Comment · Reviewer_dgoH · 2025-11-25
> >
> > Thank you for the detailed response. My concerns are partially resolved. Regarding more comprehensive numerical experiments, I suggest the followings:
> > 1. **2D Visualization:** Illustrate convergence trajectories on test functions (e.g., Rosenbrock, Beale) to provide intuition on how the damped stepsize mechanism behaves.
> > 2. **Neural Network Benchmarks:** Following the setting in "Betty Shea and Mark Schmidt. Don't Be So Positive: Negative Step Sizes in Second-Order Methods, 2024" to train neural networks on standard datasets. Compare your method against GD (to demonstrate second-order convergence) and L-BFGS (to evaluate damped stepsize vs. line search), reporting loss curves against both wall-clock time and iterations.
> > 3. **Lennard-Jones Problem:** Evaluate the method on energy minimization for Lennard-Jones clusters with $N > 100$ atoms, where vanilla Newton struggles but Newton-CG succeeds. This would demonstrate the utility in practical scenarios.

---

> ### Author Response · Authors · 2025-11-26
>
> Thank you for suggesting the experiments to support the paper. We added all three topics to the Appendix:
>
> **2D Visualization**: Newton and Damped Newton method with stepsizes, inherited from the loss transformation on Beale and Goldstein-Price funcions.
>
> **Lennard-Jones Problem**: Comparing Newton to Newton-MINRES and Newton-MINRES with loss transformations for $N=300$ atoms.
>
> **Neural Netowrk Benchmarks**: Followed the setting from "Betty Shea and Mark Schmidt. Don't Be So Positive: Negative Step Sizes in Second-Order Methods, 2024", compared SR-1 and L-BFGS with transformations to methods from the paper with and without linesearch.
>
> Please let us know if there are any remaining concerns. We would appreciate it if the additional clarifications and strengthened empirical results could motivate a re-evaluation of the submission.

---

### Official Review · Reviewer_VjE5 · 2025-11-01

**Soundness:** 3
**Presentation:** 3
**Contribution:** 3
**Rating:** 6
**Confidence:** 3

**Summary:**

This paper extends the convergence analysis of Newton’s method to pseudoconvex functions (possibly nonconvex) by composing the function $f$ with a ‘convexifier’ $\phi$ so that $\phi \circ f$ is convex. Applying Newton’s method for the composed, convex function corresponds to Newton’s method for the original nonconvex function with an adjusted step size, i.e., we can leverage the theoretical guarantees for convex functions to the nonconvex case, and such a convexifier always exists for strictly pseudoconvex functions.

**Strengths:**

- The writing is clear and easy to read overall.
- The paper includes numerical experiments that support the theoretical results.

**Weaknesses:**

See **Questions.**

**Questions:**

- To run the proposed algorithm in practical situations, the only constructive way suggested here seems to be finding the function $h$ and then using some $\phi$ that satisfies the inequality in Theorem 3. Can you explain a bit more detail on how this could be done given an arbitrary pseudoconvex functions? I would like to know how much computation will the process of computing $r(x)$ and then $\phi$ via Theorem 2 and 3 require, and whether it is negligible enough to consider doing all this to find the right step size schedules to run Newton.
- Is there a possibility that we might have slow convergence guarantees (compared to real performance) in cases when the inequality in Theorem 3 is loose? (I understand that the focus of this paper is that the convergence guarantees for pseudoconvex functions are *new*, but I am wondering if there could be cases where a better choice of $\phi$ can better capture the real dynamics or lead to a better step-size scheduling.)
- For the star-convex case, is it also possible to convert cubic Newton for the star-convexified  loss into a step-size scheduled version of cubic Newton for the original function? Or are there technical difficulties in doing the same transformation as classical Newton here?

**Typos?**

Page 4. $f \circ \phi \rightarrow \phi \circ f$

Theorem 4. I think $g$ is supposed to be star-convex (instead of convex).

---

> ### Author Response · Authors · 2025-11-24
>
> We are thankful to the reviewer for their detailed assessment and thoughtful comments. We are ready to answer the posed questions about the class of loss invariant methods and the practical aspects of implementation.
>
> 1)Can loose bounds in Theorem 3 (exponential mapping) result in poor convergence guarantees?
>
> Indeed, the inequality in Theorem 3 might lead to different convergence rates, depending on its looseness. W.l.o.g. the key point is in analyzing the properties of the function $\phi(x) = e^{c( f(x) - f_*)}$ (Corollary 2), such as  Lipschitz constant, strongly convexity constant, and self-concordance constant. Therefore, working on a compact, these constants scale exponentially. Hence, the convergence will be slower with large $c$. However, we should additionally mention that without this mapping $\phi$, Newton method will diverge for any initialization.
>
> Consequently, loose inequality in Theorem 3 might lead to poorer convergence in comparison to less loose inequality. But it depends heavily on the initial point, the choice of function $h$ and function itself. Nevertheless, with any of these “bad choices” of $h$, the method will converge, which is not observed for vanilla Newton method.
>
> 2)How can we compute the right transform for the pseudoconvex function? Comparison this with the costs of finding step sizes for Newton method
>
> Thank you for specifying the two very important points of the paper: finding right step sizes for Newton Method and practical implementation.
>
> In the paper loss invariance for the damped Newton method, therefore, the schedulers for convex problems can be converted for the transformed objectives.
>
> Regarding a constructive way of finding the transformation $\phi$, we propose a following scheme: initialize c = 1 and check matrix $ce^{c( f(x)-f_*)}\nabla^2 f(x) + c^2 e^{c(f(x)-f_*)}\nabla f(x)\nabla f(x)^T$ for positive semi definiteness. If it is not, update c = 2 * c and repeat again. If it is, conduct a damped Newton step and divide c by 2. Hence, every iteration will be considered as a damped Newton step for the convex problem. Positive definiteness test can be done resource efficiently, with the knowledge of the inner structure: matrix $\nabla f(x)\nabla f(x)^T$ is positive semi-definite, therefore, we can utilize its kernel and range properly.
>
> The only problem in this scheme is that we might conduct iterations for different convex functions at different steps. However, if all the used damped Newton schedulers preserve the monotonic decrease of the function, then, we will converge to the minimizer $x_*$, which is common for all the used convex functions, due to the transformation mapping $\phi$.
>
>
> 3)Is it possible to convert cubic Newton for star-convexified functions into a step-size scheduled version of cubic Newton for the original Newton?
>
> The vanilla cubic Newton falls into the type of Levenberg-Marquardt regularization, as the solution
>
> $$ x^{k+1} = \operatorname{argmin}\limits_{y\in \mathbb{R}^d} \\{ f(x^k) + \langle \nabla f(x^k),y-x> + \frac{1}{2}\langle \nabla^2f(x^k)(y-x^k),y-x^k\rangle + \frac{M}{6}\|y-x^k\|^3 \\}$$
>
> can be rewritten, as
>
> $$x^{k+1} = x^k - \left(\nabla^2 f(x^k) + \frac{M}{2}\|x^{k+1}-x^k\| \cdot I\right)^{-1}\nabla f(x^k)  =x^k - \left(\nabla^2 f(x^k) +\lambda \cdot I\right)^{-1}\nabla f(x^k).$$
>
> And after the loss transformation this changes to
>
> $$y^{k+1} = y^k - \left(\nabla^2 f(y^k) + \frac{\phi_x^{’'}}{\phi_x’} \nabla f(x^k)\nabla f(x^k)^T + \frac{\lambda_{\phi}}{\phi’_x}I \right)^{-1}\nabla f(x^k),$$
>
> as Lemma 2 tells. For general ($\nabla^2 f(x^k), \nabla f(x^k)$) this identity imposes $d$ scalar equation on the single unknown $\lambda_\phi$, so it admits no solution in general.
>
> 4)Typos
>
> Thanks, we fixed them!

---

### Official Review · Reviewer_nbeN · 2025-11-01

**Soundness:** 3
**Presentation:** 3
**Contribution:** 3
**Rating:** 6
**Confidence:** 2

**Summary:**

This paper studies convergent stepsizes for the Newton method through loss transformations. First, it shows that running Newton on a transformed loss is equivalent to running Newton on the original loss with a rescaled stepsize, and for convexification and star-convexification, provides necessary and sufficient conditions for such transformations to exist. In doing so, this work extends the convergence of Newton’s method to certain nonconvex functions through rescaled stepsize scheduling. Experiments visualize regions where the scaling factor becomes negative and demonstrate expanded convergence neighborhoods on test functions.

**Strengths:**

This paper offers an interesting approach to studying the Newton method’s stepsize via loss transformations, leading to nontrivial theoretical results not covered by classical analyses. It theoretically argued for a rescaled stepsize and the existence of suitable transformations for convexification, and the experiments are appropriately conducted to study the theoretical claims and practical effect.

**Weaknesses:**

Since the theoretical results rely on pseudoconvexity and on a compact set, truly nonconvex is not covered. Also, the rescaled-stepsize theorems require knowledge minimum and the minimizer, which may be hard to compute in practice.

**Questions:**

There is a typo on line 134.

Regarding the rescaled step size in Theorems, the current formulation requires knowledge of the minimizer and the minimum value. Could this be improved using adaptive step-size strategy?

**Details Of Ethics Concerns:**

There is no ethics concerns.

---

> ### Author Response · Authors · 2025-11-24
>
> We are pleased that the reviewer found our work valuable, and we thank them for their useful comments. We are ready to answer the raised questions.
>
> 1)Compact set requirement.
>
> The motivation behind compact sets is described in the article and arises naturally from the theoretical analysis of second-order methods. Most of these analyses result in a monotone decrease of the objective function: $f(x^{k+1}) \leq f(x^k)$, and in most of these analyses the boundedness of the sublevel set $\mathcal{L} = \\{ x|f(x) \leq f(x^0)\\}$ is assumed. Hence, as the iteration of the damped Newton method will not escape this bounded set, we need to convexify the function on this compact set, instead of the whole space, as in previous works in relevant field [1, 2].
>
> 2)Pseudoconvexity requirement.
>
> As shown in the [2] and in the submitted paper, strict convexity is the sufficient and necessary condition for function convexification by the loss transformation. This is the trade-off between the class of functions and the ease of transformation, as we apply the $\phi: \mathbb{R} \to \mathbb{R}$, instead of reformulating the whole minimization problem [3].
>
> 3)Demanding the knowledge of minimum and the minimizer.
>
> The knowledge of the minimizer is not required, as proposed transformation both in Theorem 3 and Corollary 3 depend on the $f(x)$ and $f(x_*)$. To be more precise, they require to bound $f(x_*)$ from below. Therefore, as in most of the problems the bound on $f(x_*)$ is known, for example, 0, we can implement our method with this bound.
>
> 3)Typos.
>
> Thanks, we fixed them!
>
> [1] Horst, Reiner. "On the convexification of nonlinear programming problems: An applications-oriented survey." European Journal of Operational Research 15.3 (1984): 382-392.
>
> [2] Schaible, Siegfried, and Israel Zang. "On the convexifiability of pseudoconvex C2-Functions." Mathematical Programming 19.1 (1980): 289-299.
>
> [3] Xia, Yong. "A survey of hidden convex optimization." Journal of the Operations Research Society of China 8.1 (2020): 1-28.

---

> ### Comment · Reviewer_nbeN · 2025-11-26
>
> Thanks the authors for their response.
>
> Could the authors elaborate on ``as in most of the problems the bound on the bound on $f(x_*)$ is known''? Is this assumption or how do we know this bound?

---

> > ### Author Response · Authors · 2025-11-27
> >
> > We thank the Reviewer nbeN for your response.
> >
> > We assume only the existence of $x_*$, so that the function attains its minimum value, as stated in the preliminaries.
> >
> > Regarding the lower bound on $f(x_*)$in most practical optimization problems, a lower bound is known from the construction of the objective. For example, when minimizing the negative log-likelihood $-\log p_{\theta} (y|x)$, we have $-\log p_{\theta}(y|x) \geq 0$ since $p_\theta (y|x)$. In physics-motivated problems, the objective is often an energy functional with a known ground-state energy obtained from theory or prior experiments.
> >
> > Using known lower bounds on the optimal value is also standard in numerical optimization. A classical example is the Polyak step size [1]:
> >
> > $$x^{t+1} = x^t - \frac{f(x^t) - f(x_*)}{\|\nabla f(x^t)\|^2}\nabla f(x^t),$$
> >
> > which explicitly relies on knowledge of $f(x_*)$. Many modern variants and stochastic adaptations of this method also utilize such information [2, 3].
> >
> > Please, let us know if you have any concerns left.
> >
> > [1] Polyak, Boris T. "Introduction to optimization." (1987).
> >
> > [2] Hazan, Elad, and Sham Kakade. "Revisiting the Polyak step size." arXiv preprint arXiv:1905.00313 (2019).
> >
> > [3] Gower, Robert M., et al. "Analysis of an idealized stochastic Polyak method and its application to black-box model distillation." arXiv preprint arXiv:2504.01898 (2025).

---

> ### Comment · Reviewer_nbeN · 2025-11-27
>
> Thanks to the authors for the response. I will keep my score and confidence level.

---

### Official Review · Reviewer_xceV · 2025-11-02

**Soundness:** 2
**Presentation:** 1
**Contribution:** 2
**Rating:** 2
**Confidence:** 4

**Summary:**

This paper studies the damped Newton's method with the goal of extending its global convergence to a class of mildly non-convex objectives via loss function transformations. Leveraging the observation that function range transformations provide a one-to-one correspondence between a Newton step on the original objective and one on the transformed objective with a modified stepsize, the authors identify a class of functions and corresponding transformations where global convergence is ensured on the new objective. Experiments are provided in support of the theory.

**Strengths:**

The topic is relevant to the research community, and the results are meaningful for computational science and engineering domains. The loss function transformation idea, while not novel in itself, is nicely leveraged for this class of problems/methods.

**Weaknesses:**

Unfortunately, the paper has quite a few shortcomings. Broadly speaking, there's missing related literature, the presentation lacks conciseness and order, the technical part contains mistakes, and experiments are not well explained. In particular:

1. **Related work**
 	* This work falls within the broader class of approaches leveraging "hidden convexity" to establish convergence guarantees. There is a broad literature here dating from the 80's and 90's and a placement within and comparison with those approaches are missing (e.g., why/why not domain transformations, rather than range are suitable for your scenario, which other settings are served by range transformations, etc.). See [2] for an old survey, and references within [3] and [4] for more recent works.
 	* In addition, work [1] leverages both domain and range transformations to transfer convergence guarantees of Newton's method from one problem to the other. Please also see references therein.


2. **Presentation**
	* The writing is carried out negligently, with
         * many missing, inappropriate, extra or mismatched articles (e.g., lines 029, 038, 040, 042, and many others);
         * typos, mathematical or otherwise (e.g., convergence rates in lines 041, 043 should be O(k^-2) and O(k^-3); line 107, dual norm argument is $g$, not $h$; the optimum is denoted both $x^{\*}$ and $x_{\*}$ e.g., in lines 316 and 320 and others; in line 053 " not necessarily" and many others);
         * verbosity and gauche phrasing (examples throughout the paper, e.g. 087 --- "Throughout the transformation invariance [...]"; subtitle 2.1);
         * missing structural components like a conclusion-type section.

      A vocabulary and grammar correction tool (e.g., Grammarly) could greatly improve the presentation.
	* Lemma 2 is stated without a concise, mathematical description of the correspondence between the Levenberg–Marquardt hessian regularization and the stepsized Newton method. As such, this result's relevance to the topic seems tangential. What is the implication of the result? Why is it important within the present work?
	* Technical assumptions are spread throughout the text, rather than being collected in statements marked explicitly as "Assumption". E.g., a central assumption about $\phi$ being strictly increasing is stated in a **footnote** on page 3.
	* Theorem 3 and its corollary are proven as "Lemma 5" in the Appendix, which is confusing. Also, the statement proven in the appendix is an incomplete version of the statement of Theorem 3 (though the result of Theorem 3 can come out of it).


3. **Technical details**
	* The current proof of Theorem 1 is incorrect. The authors use the pseudoinverse as if it were the inverse, which leads to mistakes, since the former has different properties. Specifically, the errors are
         * Going from line 686 to 688, it seems the authors use $\mathbf{H}^\dagger \mathbf{H} = \mathbf{I}$, which does not necessarily hold for pseudoinverses. Moreover, the authors seem to also use $(\mathbf{A}\mathbf{B})^{\dagger} = \mathbf{B}^\dagger\mathbf{A}^\dagger$ which, again does not hold in general.
         * For the same reason as above, line 691 does not imply line 694
	* computation of $g$ in theorem 4 depends on the unique optimum $x^*$, and there is no discussion about this limitation.

4. **Experiments**
	* Plots in sections 4.1 and 4.2 are not explained, and one has to guess what is illustrated: what do the colors code for? Does the method converge on the modified loss? The chosen test functions are not pseudoconvex (e.g., the Goldstein-Price function has several local minima), so the theory does not apply to this experiment. To my understanding, the question was whether convergence is aided by negative stepsizes, so experiments should seek settings with convergence guarantees.
	* What is the transformation $\phi$ used for the experiments in Fig. 1? Only $f$ and $L$ are stated.


[1] Izmailov, Alexey F., and Mikhail V. Solodov. "TRANSFORMATIONS OF VARIABLES AND TRANSFORMATIONS OF EQUATIONS VIA THE PERTURBED NEWTON METHOD FRAMEWORK." (2025).

[2] Horst, Reiner. "On the convexification of nonlinear programming problems: An applications-oriented survey." European Journal of Operational Research 15.3 (1984): 382-392.

[3] Fatkhullin, Ilyas, Niao He, and Yifan Hu. "Stochastic optimization under hidden convexity." arXiv preprint arXiv:2401.00108 (2023).

[4] Xia, Yong. "A survey of hidden convex optimization." Journal of the Operations Research Society of China 8.1 (2020): 1-28.

**Questions:**

* Could you please provide examples of real-world applications where pseudoconvex functions emerge?
* Could you please discuss the possible division by zero within the modified stepsize (this is in relation to the factor $\left[1 + \frac{\phi^{\prime \prime}(f(x))}{\phi^{\prime}(f(x))} (\\|\nabla f(x)\\|_x^{*})^2 \right]^{-1}$ when $\phi^{\prime \prime}(f(x))$ is negative)?
* Could you please provide a comparison with the missing related literature mentioned above?
* Could you please motivate your choice of test functions in the experiments of section 4? Also, please explain the color coding.

---

> ### Author Response · Authors · 2025-11-24
>
> We thank the Reviewer for careful examining our paper. We updated the paper and are ready to address your concerns.
>
> 1)Unnecessary citations (Newton, Raphson)
>
> Newton and Raphson proposed the Newton method, that is widely used in numerical optimization. We mentiona their works, as consider them relevant to the topic of our manuscript.
>
> 2)Missing conclusion
>
> Due to the strict page limit we were not able to include the conclusion section and will do it with the increased page limit.
>
> 3)The importance of lemma 2
>
> General Levenberg-Marquardt regularization is written as following:
> $$ x^+ = x - \left(\nabla^2 f(x) + \lambda \cdot I\right)^{-1}\nabla f(x),$$
> and include a wide range of methods, including, for instance, gradient quadratic regularization [1] and cubic Newton [2]. Therefore, these methods are not loss invariant.
>
> 4)Question about pseudoinverse
>
> Thank you for pointing out this ambiguity. We demanded $\nabla f(x)$ to be in Range$(\nabla^2 f(x))$ specifically to derive the needed equality. This broadens the class of the applicable functions, as we do not require Hessian to be invertible.
>
> Thank you for pointing out the cornercase about possible division by zero. Indeed this should be avoided, as Sherman-Morrison formula prescribes. You can check the clarified version of the proof in the edited manuscript.
>
> 5)Comparison to the related work
>
> Indeed, there are several works, investigating the convex reformulation of the problems. However, they address slightly different questions, therefore, due to the strict page limitation we were not able to include them in the submitted version. We added the block into the appendix of the updated paper. The brief comparison is also presented below:
>
> The equivalence of some nonconvex problems and convex ones can be beneficial, since the methods for convex optimization are usually superior to nonconvex. However, these transformations are often computationally burdensome - we transform variables [3, 4], or the problems as the constraints [5, 6]. Furthermore, these analyses only claim that the optimal values of the problems will coincide. Hence, we cannot solve the nonconvex problem without the knowledge of the transformation. We, on the other side, propose simple one-dimensional transformations, that are easy to compute.
>
> 6)Plots are not explained properly
>
> Experiments’ description can be found in the corresponding section and legend. When section 4.1 investigates the sign flips in the step sizes, section 4.2 investigates convergence of the transformed losses. Different colors visualize the convergence to different optimas, as described in the legend. It can be seen that applying damped Newton method to the transformed problem may increase the convergence regions.
>
> Since our theory is derived for strictly pseudoconvex functions, it is also questionable whether loss transformations will be useful for other nonconvex functions. By these experiments we demonstrate and visualize that applying transformation  for nonconvex functions will indeed enlarge the convergence area.
>
> 7)Motivation behind the choice of functions
>
> Polynomial and logarithmic transformation were chosen for the experiments, as they simultaneously show two main properties of the loss transformation - sign flips of the step sizes and the enlarging the area of convergence.
>
> 8)Real life examples of pseudoconvex functions
>
> Several examples of pseudoconvex losses can be found in the related literature, for instance in [7]. They are widely used in robust optimization, where models need to be less influenced by outliers. The examples of pseudoconvex losses are given in the Table 2.
>
> Furthermore, theory derived for pseudoconvex optimization might be relevant to other large-scale problems, as it was shown with the convex optimization [8]. As we have mentioned earlier and shown in the manuscript, our paper aligns with the nonconvex problems, enlarging the convergence area.
>
> 9)Presentation
>
> Thanks for pointing out, we fixed the typos and reorganised the text in the updated version of the manuscript. Special attention was given to clarifying the appendix.
>
> [1] Doikov, Nikita, and Yurii Nesterov. "Gradient regularization of Newton method with Bregman distances."
>
> [2] Nesterov, Yurii, and Boris T. Polyak. "Cubic regularization of Newton method and its global performance."
>
> [3] Fatkhullin, Ilyas, Niao He, and Yifan Hu. "Stochastic optimization under hidden convexity."
>
> [4] Izmailov, Alexey F., and Mikhail V. Solodov. "TRANSFORMATIONS OF VARIABLES AND TRANSFORMATIONS OF EQUATIONS VIA THE PERTURBED NEWTON METHOD FRAMEWORK." (2025).
>
> [5] Xia, Yong. "A survey of hidden convex optimization."
>
> [6] Horst, Reiner. "On the convexification of nonlinear programming problems: An applications-oriented survey."
>
> [7] Barron, Jonathan T. "A general and adaptive robust loss function."
>
> [8] Schaipp, Fabian, et al. "The Surprising Agreement Between Convex Optimization Theory and Learning-Rate Scheduling for Large Model Training."

---

### Author Response · Authors · 2025-11-26
**Additional experimental validation**

We thank Reviewer dgoH for suggesting additional experiments that strengthen our manuscript:

**Additional 2D Visualization** on benchmark non-convex functions. We demonstrate how negative step sizes inherited from the loss transformation lead to convergence.

**Analyzing the Lennard–Jones** potential energy minimization using Newton–MINRES with loss transformations.

**Neural Network** LibSVM benchmarks. Step sizes inherited from loss transformations are more effective than strong Wolfe line searches.

All experiments are presented in the Appendix.

We invite all Reviewers to join the discussion regarding the additional experimental validation and the theoretical aspects of our manuscript.

---

### Author Response · Authors · 2025-12-03
**Discussion Summary by Authors**

Dear AC and Reviewers,

We thank you for your time and effort. We would like to provide a brief summary of the discussion during the review and rebuttal periods. First of all, we highlight the positive comments about our paper, which demonstrate that the Reviewers invested significant effort in evaluating our work and acknowledged its intended contributions:

> Reviewer **xceV**: The topic is relevant to the research community, and the results are meaningful for computational science and engineering domains. The loss function transformation idea, while not novel in itself, is nicely leveraged for this class of problems/methods.

> Reviewer **nbeN**: This paper offers an interesting approach to studying the Newton method’s stepsize via loss transformations, leading to nontrivial theoretical results not covered by classical analyses.

> Reviewer **VjE5**: Applying Newton’s method for the composed, convex function corresponds to Newton’s method for the original nonconvex function with an adjusted step size, i.e., we can leverage the theoretical guarantees for convex functions to the nonconvex case, and such a convexifier always exists for strictly pseudoconvex functions.

> Reviewer **dgoH**: The motivations for using loss transformation and stepsize scheduling are clear, and the writing flows smoothly, from the discussion on convex and pseudoconvex problems to the convexification of a broad class of nonconvex problems.

---
Below, we summarize the questions that emerged during the discussion.

Reviewer **xceV** raised important concerns about the proof of Theorem 1, in particular regarding the behavior of the pseudoinverses. In our official comment and in the updated PDF we deliberately expanded this section, explicitly mentioning the Hessian, which is standard for Newton–CG. Therefore, we have theoretically justified applying our method to nonconvex problems, and to pseudoconvex problems in particular. The concise proof can be found in the appendix of the submitted manuscript.

Among other concerns were the lack of related literature, which we have now added to the appendix, as well as the quality of the presentation. During the rebuttal process we corrected the mentioned imprecisions. We believe we have addressed all the concerns raised, and we are frustrated that no further discussion followed due to this unfortunate incident.

The other Reviewers did not specifically question the presentation quality, but posed other important questions. Reviewer **nbeN** was concerned about the requirement of knowing the minimum value of the objective. After a brief discussion, they decided to keep their score (6) with confidence level 2.

Reviewer **Vje5** was particularly interested in practical implementations of our method, as well as in expanding the class of loss-invariant second-order methods. In our response we elaborated on practical aspects of the implementation and discussed the vanilla cubic Newton method from the Levenberg–Marquardt regularization point of view.

Reviewer **dgoH** also questioned the loss-invariance property of other second-order methods, which we addressed thoroughly. They further requested a more elaborate experimental section, validating the benefits of the loss transforms. We want to thank this reviewer for engaging in the discussion and suggesting experimental setups. We added all proposed experiments and indeed demonstrated that applying loss transformations not only increases the region of convergence, but also leads to performance improvements across a wide range of tasks. It is unfortunate that the discussion ended, we believe that these results, and any results from further suggested setups, would positively influence the final score.

---
Thank you once again for the valuable feedback. We believe we addressed all the posed questions.

---

### Meta-Review · Area_Chair_LdCT · 2026-01-04

**Summary:**

This paper studies the damped Newton's method with the goal of extending its global convergence to a class of mildly non-convex objectives via loss function transformations. Two reviewers gave positive scores, and one reviewer has concerns about soundness and experimental results. Besides, Reviewer xceV with high confidence was against acceptance from a careful proofreading, which is mainly due to the missing related literature, the presentation, the technical mistakes, and the experiments. From my own reading, I agree with Reviewer xceV that some shortcomings cannot be well addressed in revisions.

**Reviewer Concerns:**

It may need extra time and effort to verify the new revision, which is revised from one concern of Reviewer xceV: "The current proof of Theorem 1 is incorrect."

**Reviewer Scores:**

I appreciate the authors' efforts to address the concerns raised in the rebuttal, and I think Reviewer xceV might increase their score.

---

### Decision · Program_Chairs · 2026-01-26

Reject